# A Theoretical Study of Inductive Biases in Contrastive Learning

**Jeff Z. HaoChen & Tengyu Ma**
Department of Computer Science
Stanford University
{jhaochen,tengyuma}@stanford.edu

## Abstract

Understanding self-supervised learning is important but challenging. Previous theoretical works study the role of pretraining losses, and view neural networks as general black boxes. However, the recent work of Saunshi et al. (2022) argues that the model architecture — a component largely ignored by previous works — also has significant influences on the downstream performance of self-supervised learning. In this work, we provide the first theoretical analysis of self-supervised learning that incorporates the effect of inductive biases originating from the model class. In particular, we focus on contrastive learning — a popular self-supervised learning method that is widely used in the vision domain. We show that when the model has limited capacity, contrastive representations would recover certain special clustering structures that are compatible with the model architecture, but ignore many other clustering structures in the data distribution. As a result, our theory can capture the more realistic setting where contrastive representations have much lower dimensionality than the number of clusters in the data distribution. We instantiate our theory on several synthetic data distributions, and provide empirical evidence to support the theory.

## 1 Introduction

Recent years have witnessed the effectiveness of pre-trained representations, which are learned on unlabeled data with self-supervised losses and then adapted to a wide range of downstream tasks (Chen et al., 2020a;b; He et al., 2020; Caron et al., 2020; Chen et al., 2020c; Gao et al., 2021; Su et al., 2021; Chen & He, 2020; Brown et al., 2020; Radford et al., 2019). However, understanding the empirical success of this emergent pre-training paradigm is still challenging. It requires novel mathematical frameworks and analyses beyond the classical statistical learning theory. The prevalent use of deep neural networks in self-supervised learning also adds to the mystery.

Many theoretical works focus on isolating the roles of self-supervised losses, showing that they encourage the representations to capture certain structures of the unlabeled data that are helpful for downstream tasks (Arora et al., 2019; HaoChen et al., 2021; 2022; Wei et al., 2021; Xie et al., 2021; Saunshi et al., 2020). However, these works oftentimes operate in the sufficient pre-training data (polynomial in the dimensionality) or even infinite pre-training data regime, and view the neural network as a *black box*. The only relevant property of neural networks in these works is that they form a parameterized model class with finite complexity measure (e.g., Rademacher complexity).

Recently, Saunshi et al. (2022) argue that the pre-training loss is *not* the only contributor to the performance of self-supervised learning, and that previous works which view neural networks as a black box cannot tell apart the differences in downstream performance between architectures (e.g., ResNet (He et al., 2015) vs vision transformers (Dosovitskiy et al., 2020)). Furthermore, self-supervised learning with an appropriate architecture can possibly work under more general conditions and/or with fewer pre-training data than predicted by these results on general architecture. Therefore, a more comprehensive and realistic theory needs to take into consideration the inductive biases of architecture.

This paper provides the first theoretical analyses of the inductive biases of *nonlinear* architectures in self-supervised learning. Our theory follows the setup of the recent work by HaoChen et al. (2021)

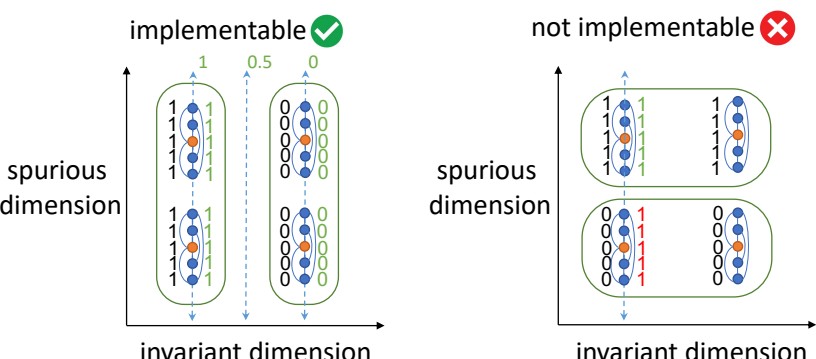

Figure 1: **A simple example where the linear function class learns the correct feature and ignores the spurious feature.** A simplified version of the synthetic example proposed in Saunshi et al. (2022). The orange points are the original data and blue points are augmented data (obtained by adding noise in the spurious dimension). The dimension invariant to augmentation is desired. Edges represent positive pairs that are constructed from augmentation. We say a real-valued function *implements* a cluster if it outputs 1 on the cluster and outputs 0 on all other data. We note that here implementing means matching the *exact* value, rather than simply matching the label after applying some linear threshold. The figure above shows two possible ways to partition the data into two clusters, but only the one on the left-hand side (which captures the invariant dimension) is implementable by a linear function. Here we use black numbers to indicate the target output on the data, and green numbers to indicate the output of the implementing function which extrapolates outside of the data support. Note that linear model is *not* composed with a threshold function. The partition on the right hand side is not implementable because any linear model that outputs constant 1 on the upper-left small cluster would also output 1 on the bottom-left small cluster due to linear extrapolation. Here we use red numbers to indicate the output of the linear function that contradicts with the target.

on contrastive learning and can be seen as a refinement of their results by further characterizing the model architecture's impact on the learned representations.

We recall that HaoChen et al. (2021) shows that contrastive learning, with sufficient data and a parameterized model class of finite complexity, is equivalent to spectral clustering on a so-called *population positive-pair graph*, where nodes are augmented images and an edge between the nodes $x$ and $x'$ is weighted according to the probability of encountering $(x, x')$ as a positive pair. They essentially assume that the positive-pair graph contains several major semantically-meaningful clusters, and prove that contrastive representations exhibit a corresponding clustering structure in the Euclidean space, that is, images with relatively small graph distance have nearby representations.

Their results highly rely on the clustering property of the graph—the representation dimensionality and pre-training sample complexity both scale in the number of clusters. The important recent work of Saunshi et al. (2022), however, demonstrates with a synthetic setting that contrastive learning can provably work with linear model architectures even if the number of clusters is huge (e.g., exponential in the dimensionality). Beyond the simple synthetic example discussed in their paper, there has been no previous work that formally characterizes this effect in a general setting.

In this work, we develop a general theory that leverages the inductive bias to avoid the dependency on the potentially huge number of clusters: although there exists a large number of clusters in the positive-pair graph, the number of clusters *implementable by the model* (which we call *minimal implementable clusters*) could be much smaller, even exponentially. Figure 1 shows an example where a linear function can only implement one clustering structure but not the other, despite both being valid clusters in the positive-pair graph. It's possible that a minimal implementable cluster consists of multiple well-separated sub-clusters but none of these sub-clusters can be implemented by the model class.

We show that contrastive representations would only recover the clustering structures that are compatible with the model class, hence low-dimensional contrastive learned representations would work well on the downstream tasks. Concretely, suppose the number of minimal implementable clusters is $m$ which can be much smaller than the number of natural clusters in the graph $\tilde{m}$. HaoChen et al. (2021) prove the efficacy of contrastive learning assuming the representation dimensionality (hence also sample complexity) is larger than $\tilde{m}$. We develop a new theory (Theorem 1) that makes the representation dimensionality only depend on $m$ instead of $\tilde{m}$. We also extend this result to a more complex setting where we can deal with even more structured clusters, e.g., when there are $2^s$ clusters with certain geometric structures, but the representation dimensionality can scale with only $s$ instead of $2^s$. See Theorem 2 and its instantiation on Example 1 for this result.

We instantiate our theory on several synthetic data distributions and show that contrastive learning with appropriate model architectures can reduce the representation dimensionality, allowing better sample complexity. We consider a data distribution on a hypercube first proposed by Saunshi et al. (2022) which contains a small subspace of features that are invariant to data augmentation and a large subspace of spurious features. When the function class is linear, we show that the contrastive representations can solve downstream binary classification tasks if the downstream label only depends on one dimension of invariant features (Theorem 3). When the function class is ReLU networks (hence more expressive), we show that the contrastive representations can solve more diverse downstream classification problems where the label can depend on all invariant features (Theorem 4). We also provide examples for Lipschitz-continuous function classes (Theorem 5) and convolutional neural networks (Theorem 6).

We provide experimental results to support our theory. We propose a method to test the number of implementable clusters of ResNet-18 on the CIFAR-10 dataset and show that there are indeed only a small number of implementable clusters under the model architecture constraint (Section B).

## 2 RELATED WORKS

The empirical success of contrastive learning has attracted a series of theoretical works that study the contrastive loss (Arora et al., 2019; HaoChen et al., 2021; 2022; Tosh et al., 2020; 2021; Lee et al., 2020; Wang et al., 2021; Nozawa & Sato, 2021; Ash et al., 2022; Tian, 2022), most of which treat the model class as a black box except for Lee et al. (2020) which studies the learned representation with linear models, and Tian (2022) and Wen & Li (2021) which study the training dynamics of contrastive learning for linear and 2-layer ReLU networks.

Several theoretical works also study non-contrastive methods for self-supervised representation learning (Wen & Li, 2022; Tian et al., 2021; Garrido et al., 2022; Balestriero & LeCun, 2022). There are also works theoretically studying self-supervised learning in other domains such as language modeling (Wei et al., 2021; Xie et al., 2021; Saunshi et al., 2020).

## 3 FROM CLUSTERS TO MINIMAL IMPLEMENTABLE CLUSTERS

In this section, we introduce our main theoretical results regarding the role of inductive biases of architectures in contrastive learning. Recall that contrastive learning encourages two different views of the same input (also called a *positive pair*) to have similar representations, while two random views of two different inputs (also called a *negative pair*) have representations that are far from each other. Formally, we use $p_{\text{data}}$ to denote the distribution of a random view of random input, use $p_{\text{pos}}$ to denote the distribution of a random positive pair, and $\mathcal{X}$ to denote the support of $p_{\text{data}}$. For instance, $\mathcal{X}$ is the set of all augmentations of all images for visual representation learning.

Following the setup of HaoChen et al. (2022), for a representation map $f : \mathcal{X} \to \mathbb{R}^k$ where $k$ is the representation dimensionality, we learn the contrastive representation by minimizing the following generalized spectral contrastive loss:

$$\mathcal{L}_\lambda(f) := \mathbb{E}_{(x,x+)\sim p_{\text{pos}}}[\|f(x) - f(x^+)\|_2^2] + \lambda \cdot R(f), \tag{1}$$

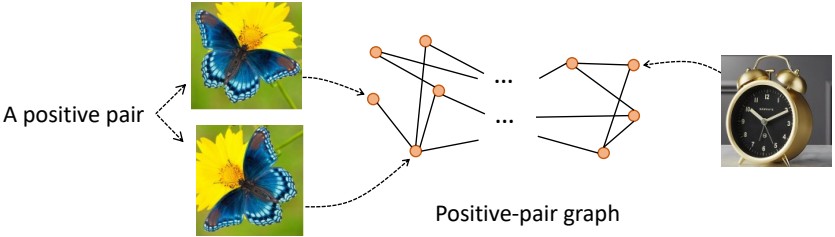

Figure 2: **A demonstration of the positive-pair graph.** When the positive pairs are formed by applying data augmentation (such as rotation) to the same natural image, data with the same semantic meaning (e.g., the two butterfly images) tend to belong to the same cluster in the positive-pair graph. Datapoints with different semantic meanings (e.g., a butterfly image and a clock image) would not be connected in the positive-pair graph, hence belongs to different clusters.

where $\lambda > 0$ is a hyperparameter indicating the regularization strength, and the regularizer normalizes the representation covariance towards the identity matrix:

$$R(f) := \left\| \mathbb{E}_{x \sim p_{\text{data}}}[f(x)f(x)^\top] - \mathbb{I} \right\|_F^2. \tag{2}$$

This loss is very similar to the popular Barlow Twins loss (Zbontar et al., 2021) and has been shown to empirically work well (HaoChen et al., 2021).

Theoretically, the prior work proposes the notion of *positive-pair graph* with $\mathcal{X}$ being the vertex set and an edge between the nodes $x$ and $x'$ is weighted according to the probability of encountering $(x, x')$ as a positive pair (i.e., $p_{\text{pos}}(x, x')$). This graph is defined on the **population** data, and intuitively captures the semantic relationship between different data — when the positive pairs are formed by applying data augmentation to the same natural data, it is expected that datapoints in the same cluster in the positive-pair graph would have similar semantic meanings. Figure 3 gives a demonstration of the positive-pair graph.

Their analysis shows that learning contrastive representations with the above loss is equivalent to spectral clustering Ng et al. (2001) on this positive-pair graph, hence can learn meaningful representations when the graph has clustering structures.

Different from the prior work, we study the representation map that minimizes the contrastive loss *within a certain function class* $\mathcal{F}$. Here we assume functions in $\mathcal{F}$ map data in $\mathcal{X}$ to representations in $\mathbb{R}^k$ for some dimensionality $k$. The main contribution of our result is the improvement of $k$ due to the consideration of this specific function class: by studying the representation learned within a constrained model class $\mathcal{F}$, we will show that the necessary representation dimensionality $k$ is much smaller than that required in the prior work. As a result, the sample complexity for the downstream labeled task would be improved compared to the prior work.

Let $\{S_1, S_2, \cdots, S_m\}$ be a $m$-way partition of $\mathcal{X}$, i.e., they are disjoint non-empty subsets of $\mathcal{X}$ such that $\mathcal{X} = \cup_{i \in [m]} S_i$. For any $x \in \mathcal{X}$, let $\text{id}_x$ be the index such that $x \in S_{\text{id}_x}$. We consider a partition of the graph such that there is not much connection between any two clusters, which is formalized by the following assumption.

**Assumption 1** ($\alpha$-separability). *The probability of a positive pair belonging to two different sets is less than $\alpha$:*

$$\Pr_{(x, x^+) \sim p_{\text{pos}}} (\text{id}_x \neq \text{id}_{x^+}) \leq \alpha. \tag{3}$$

We consider downstream tasks that are $r$-way classification problems with label function $y(\cdot) : \mathcal{X} \to [r]$. We assume that the downstream task aligns with the clusters:

**Assumption 2.** *The downstream label $y(x)$ is a constant on each $S_i$.*

Our key assumptions about the function class are that it can implement desirable clustering structures (Assumption 4) but cannot break the positive-pair graph into too many clusters (Assumption 3).

Let $S \subset \mathcal{X}$ be a subset of $\mathcal{X}$, $p_{\text{data}}^S$ be the distribution $p_{\text{data}}$ restricted to set $S$, and $p_{\text{pos}}^S$ be the positive pair distribution $p_{\text{pos}}$ conditioned on both datapoints in the pair belonging to set $S$. For any

function $g : S \rightarrow \mathbb{R}$, we define the following expansion quantity:

$$Q_S(g) := \frac{\mathbb{E}_{(x,x^+) \sim p_{\text{pos}}^S}[(g(x) - g(x^+))^2]}{\mathbb{E}_{x \sim p_{\text{data}}^S, x' \sim p_{\text{data}}^S}[(g(x) - g(x'))^2]}. \quad (4)$$

We let $Q_S(g) = \infty$ if the denominator is 0. Here the numerator represents the discrepancy between a random positive pair, and the denominator represents the global variance of $g$. Intuitively, a smaller value $Q_S(g)$ means that function $g$ does a better job at separating the set $S$ into disjoint sub-clusters, and hence implements an inner-cluster connection structure that is sparse. For instance, if $S$ contains two disjoint sub-clusters, and $g$ has different constant values on each of them, then $Q_S(g) = 0$. On the other hand, if $S$ is densely connected, then $Q_S(g) > 0$ regardless of the choice of $g$.

The first assumption about the function class $\mathcal{F}$ assumes that no function in the class can break one cluster into two well-separated sub-clusters:

**Assumption 3** ($\mathcal{F}$-implementable inner-cluster connection larger than $\beta$). *For any function $f \in \mathcal{F}$ and any linear head $w \in \mathbb{R}^k$, let function $g(x) = w^\top f(x)$. For any $i \in [m]$ we have that:*

$$Q_{S_i}(g) \geq \beta. \quad (5)$$

We note that when the function class $\mathcal{F}$ contains *all* the functions from $\mathcal{X}$ to $\mathbb{R}^k$, Assumption 3 essentially says that each of $\{S_1, S_2, \cdots, S_m\}$ has large internal expansion, hence recovers Assumption 3.5 in HaoChen et al. (2021). However, when $\mathcal{F}$ has limited capacity, each cluster $S_i$ can still contain well-separated sub-clusters, but just those sub-clusters cannot be implemented by functions in $\mathcal{F}$.

Assumption 3 implies that the function class cannot be too expressive. However, in order for the learned representation map to be useful for downstream tasks, it needs to be expressive enough to represent the useful information. Thus, we introduce the following assumption on the function class.

**Assumption 4** (Implementability). *Recall that $\text{id}_x$ is the index such that $x \in S_{\text{id}_x}$. There exists a function $f \in \mathcal{F}$ such that $f(x) = e_{\text{id}_x}$ for all $x \in p_{\text{data}}$ where $e_i \in \mathbb{R}^m$ is the vector where the $i$-th dimension is 1 and other dimensions are 0.*

When both Assumption 3 and Assumption 4 hold, we say $\{S_1, S_2, \cdots, S_m\}$ are *minimal implementable clusters* with respect to $\mathcal{F}$.

We also introduce the following Assumption 5 which is true for any function class implemented by a neural network where the last layer is linear. We note that this assumption is needed only for the technical rigour of the proof, and is not essential to the conceptual message of our theory.

**Assumption 5** (Closure under scaling). *For any function $f \in \mathcal{F}$ and vector $u \in \mathbb{R}^m$, define function $f'(x) = u \odot f(x)$ where $\odot$ means element-wise product. Then, we have $f' \in \mathcal{F}$.*

Let $P_{\min} := \min_{i \in [m]} \Pr_{x \sim p_{\text{data}}}(x \in S_i)$ and $P_{\max} := \max_{i \in [m]} \Pr_{x \sim p_{\text{data}}}(x \in S_i)$ be the sizes of the smallest and largest sets respectively. Under the above assumptions, we have the following theorem that shows learning a representation map within $\mathcal{F}$ and representation dimensionality $k = m$ can solve the downstream task:

**Theorem 1.** *Suppose $\{S_1, S_2, \cdots, S_m\}$ are minimal implementable clusters with respect to $\mathcal{F}$ (i.e., Assumptions 1 and 3 hold), and the function class $\mathcal{F}$ satisfies Assumptions 4 and 5. For $\lambda > \alpha/P_{min}$, consider a learned representation map $\hat{f} = \arg\min_{f \in F} \mathcal{L}_\lambda(f)$ that minimizes the contrastive loss. Then, when $k = m$, for any downstream task that satisfies Assumption 2, there exists a linear head $W \in \mathbb{R}^{r \times k}$ which achieves downstream error*

$$\mathbb{E}_{x \sim p_{\text{data}}}\left[\|W\hat{f}(x) - e_{y(x)}\|_2^2\right] \leq \frac{\alpha}{\beta} \cdot \frac{P_{max}}{P_{min} - \alpha}. \quad (6)$$

We note that $P_{\max} \approx P_{\min}$ when the partitions are balanced. Thus, so long as $\alpha \ll P_{\min}$ (i.e., the probability of a positive pair crossing different clusters is smaller than the probability of it containing data from the smallest cluster), the right-hand side is roughly $\alpha/\beta$. Thus, when the inter-cluster connection $\alpha$ is smaller than the inner-cluster connection that is implementable by the function class $\beta$, the downstream accuracy would be high.

**Comparison with HaoChen et al. (2021).** We note that our result requires $k = m$, whereas HaoChen et al. (2021) provides analysis in a more general setting for arbitrary $k$ that is large enough. Thus, when the function class $\mathcal{F}$ is the set of all functions, our theorem recovers a special case of HaoChen et al. (2021). Our result requires a stricter choice of $k$ mainly because when $\mathcal{F}$ has limited capacity, a higher dimensional feature may contain a lot of "wrong features" while omitting the "right features", which is a phenomenon that doesn't occur when $\mathcal{F}$ contains all functions.

## 4 AN EIGENFUNCTION VIEWPOINT

In this section, we introduce an eigenfunction perspective that generalizes the theory in the previous section to more general settings. We first introduce the background on eigenfunctions and discuss its relation with contrastive learning. Then we develop a theory that incorporates the model architecture with assumptions stated using the language of eigenfunctions. The advantage over the previous section is that we can further reduce the required representation dimensionality when the minimal implementable clusters exhibit certain internal structures.

Our theory relies on the notion of *Laplacian operator* $\mathbb{L}$ which maps a function $g : \mathcal{X} \to \mathbb{R}$ to another function $\mathbb{L}(g) : \mathcal{X} \to \mathbb{R}$ defined as follows.

$$\mathbb{L}(g)(x) := g(x) - \int \frac{p_{\text{pos}}(x, x')}{p_{\text{data}}(x)} g(x') dx'. \tag{7}$$

We say a function $g$ is an eigenfunction of $\mathbb{L}$ with eigenvalue $\psi \in \mathbb{R}$ if for some scalar $\psi$

$$\mathbb{E}_{x \sim p_{\text{data}}} \left[ (\psi \cdot g(x) - \mathbb{L}(g)(x))^2 \right] = 0. \tag{8}$$

This essentially means that $L(g) = \psi \cdot g$ on the support of $p_{\text{data}}$. Intuitively, small eigenfunctions (i.e., eigenfunctions with small eigenvalues) correspond to clusters in the positive-pair graph. To see this, let $g$ implement the indicator function of cluster $S$, i.e., $g(x) = 1$ if $x \in S$, and $g(x) = 0$ if $x \notin S$. One can verify that $\mathbb{L}(g)(x) = 0$ for all $x$, thus $g$ is an eigenfunction with eigenvalue $0$.

In this section, we provide a generalized theory based on characterizing the implementability of eigenfunctions. Intuitively, we will assume that there exists $k$ (and only $k$) orthogonal eigenfunctions in the function class with very small eigenvalue, and the downstream task can be solved by these eigenfunctions. More formally, let $\phi \geq 0$ be a very small real number (can be thought as $0$), and $f_{\text{eig}}(x) : \mathcal{X} \to \mathbb{R}^k$ be a $k$-dimensional representation map in the function class $\mathcal{F}$ such that

$$\mathbb{E}_{(x,x^+) \sim p_{\text{pos}}} \left[ \left\| f_{\text{eig}}(x) - f_{\text{eig}}(x^+) \right\|_2^2 \right] \leq \phi \tag{9}$$

and

$$\mathbb{E}_{x \sim p_{\text{data}}} \left[ f_{\text{eig}}(x) f_{\text{eig}}(x)^\top \right] = \mathbb{I}. \tag{10}$$

Intuitively, when $\phi$ is small, each dimension of $f_{\text{eig}}$ corresponds to one eigenfunction of the graph Laplacian with small eigenvalue, as formalized by the following Proposition 1 in the case of $\phi = 0$.

**Proposition 1.** *For any $i \in [k]$ and any function $f_{eig}$ satisfying equation 9 with $\phi = 0$ and equation 10, we have that function $g(x) = f_{eig}(x)_i$ is an eigenfunction of $\mathbb{L}$ with eigenvalue $0$.*

Recall that our assumptions in the previous section intuitively say that even though a larger number of clusters exist in the positive-pair graph, many of them are not implementable by the function class. From the eigenfunction viewpoint, this means that only a small number of eigenfunctions with small eigenvalue are in the function class. Thus, we can make the following corresponding assumption which says that the vector-valued function $f_{\text{eig}}$ contains all the implementable eigenfunctions with small eigenvalue.

**Assumption 6.** *Suppose $g$ is a function implementable by $\mathcal{F}$ (in the sense that $g(x) = f(x)_i$ for some $f \in \mathcal{F}$ and $i \in [k]$) and*

$$\mathbb{E}_{(x,x^+) \sim p_{\text{pos}}}[(g(x) - g(x^+))^2] \leq \tilde{\phi} \cdot \mathbb{E}_{x \sim p_{\text{data}}} \left[ g(x)^2 \right], \tag{11}$$

*then there exists $\tilde{w} \in \mathbb{R}^k$ such that*

$$\mathbb{E}_{x \sim p_{\text{data}}} \left[ (\tilde{w}^\top f_{eig}(x) - g(x))^2 \right] \leq \epsilon. \tag{12}$$

*Here both $\tilde{\phi}$ and $\epsilon$ are very small and can be thought as $0$.*

We consider downtream tasks that can be solved by $f_{\text{eig}}$. Let $\vec{y}(x) \in \mathbb{R}^r$ be a vector that represents the downstream label of data $x$ (e.g., the one-hot embedding of the label when the downstream task is classification). We have the following assumption on the downstream task:

**Assumption 7.** *There exists a linear head $W^* \in \mathbb{R}^{r \times m}$ with norm $\|W^*\|_F \leq B$ such that*

$$\mathbb{E}_{x \sim p_{\text{data}}}\left[\left\|W^* f_{eig}(x) - \vec{y}(x)\right\|_2^2\right] \leq \zeta. \tag{13}$$

*Here $\zeta$ is very small and can be thougth as $0$.*

We have the following theorem using the above two assumptions:

**Theorem 2.** *Suppose function $f_{eig} \in \mathcal{F}$ satisfies Assumptions 6 with $(\tilde{\phi}, \epsilon)$ and Assumption 7 with $(B, \zeta)$. Suppose $\tilde{\phi} > \phi$ or $\tilde{\phi} = \phi = 0$. Then, for any $\lambda > 0$ such that $\phi \leq \tilde{\phi}(1 - \sqrt{\phi/\lambda})$ and learned representation map $\hat{f} = \arg\min_{f \in \mathcal{F}} \mathcal{L}_\lambda(f)$, there exists a linear head $W \in \mathbb{R}^{r \times k}$ such that*

$$\mathbb{E}_{x \sim p_{\text{data}}}\left[\left\|W\hat{f}(x) - \vec{y}(x)\right\|_2^2\right] \lesssim \zeta + B^2 k\left(\epsilon + \frac{\phi}{\lambda}\right). \tag{14}$$

Since $\zeta$, $\epsilon$ and $\phi$ are all very small values, the RHS of equation 14 is very small, hence the learned representation acheives small downstream error. As we will see in the first example in the next section, Theorem 2 indeed allows the representation dimensionality to be smaller than the number of minimal implementable clusters in the graph, hence generalizes the result in the previous section.

**Relationship between Theorem 2 and Theorem 1**. We note that in the setting of Theorem 1, the identity function of each minimal implementable cluster would be an achievable eigenfunction. Theorem 2 considers a more general situation than Theorem 1 where the minimal implementable clusters may not be well-defined, yet still we can show good results when the dimensionality is equal to the number of achievable eigenfunctions. We will mainly use Theorem 2 for the examples because it's more general and easier to be used, whereas we present Theorem 1 because it's more intuitive to understand. For instance, in our Example 1, Theorem 1 only applies when $s = 1$, whereas Theorem 2 applies for arbitrary $s$.

## 5 INSTANTIATIONS ON SEVERAL SYNTHETIC DATA DISTRIBUTIONS

In this section, we instantiate our previous theory on several examples of data distributions and show that when the model class has limited capacity, one can learn low-dimensional representations using contrastive learning and solve the downstream task with simple linear probing. In all of these examples, if we use a much more expressive model class, the representation dimensionality needs to be much higher, and hence more downstream samples are needed. These results demonstrate the benefit of leveraging inductive biases of the model architecture in contrastive learning.

### 5.1 LINEAR FUNCTIONS

Our first example is the hypercube example proposed in Saunshi et al. (2022).

**Example 1.** *The natural data $\bar{x} \sim \{-1, 1\}^d$ is the uniform distribution over the $d$-dimensional cube. Given a natural data $\bar{x}$, an augmented data $x \sim \mathcal{A}(\bar{x})$ is sampled as follows: first uniformly sample a scalar $\tau \sim [\frac{1}{2}, 1]$, then scale the $(s+1)$-th to $d$-th dimensions of $\bar{x}$ with $\tau$, while keeping the first $s$ dimensions the same. Intuitively, the last $d - s$ dimensions correspond to spurious features that can be changed by data augmentation, and the first $s$ dimensions are invariance features that contain information about the downstream task. The downstream task is a binary classification problem, where the label $y(x) = sgn(x_i)$ is the sign function of one of the first $s$ dimensions $i \in [s]$.*

We consider contrastive learning with the linear function class defined below:

**Definition 1** (Linear function class). *Let $U \in \mathbb{R}^{k \times d}$ be a matrix and we use $f_U(x) = Ux$ to denote the linear function with weight matrix $U$. We define the $k$-dimensional linear function class as $\mathcal{F}_{\text{linear}} = \{f_U : U \in \mathbb{R}^{k \times d}\}$.*

Saunshi et al. (2022) directly compute the learned representations from contrastive learning. Here we show that the example can be viewed as an instantiation of our more general Theorem 2. In particular, we have the following result:

**Theorem 3.** *In Example 1, suppose we set the output dimensionality as $k = s$ and learn a linear representation map that minimizes the contrastive loss $\hat{f} = \arg\min_{f \in \mathcal{F}_{\text{linear}}} \mathcal{L}_\lambda(f)$ for any $\lambda > 0$. Then, there exists a linear head $w \in \mathbb{R}^k$ such that*

$$\mathbb{E}_{x \sim p_{\text{data}}}[(w^\top \hat{f}(x) - y(x))^2] = 0. \tag{15}$$

*In contrast, suppose the function class is the set of universal function approximators $\mathcal{F}_{\text{uni}}$. So long as the output dimensionality is no more than $2^{d-1}$, there exists solution $\hat{f}' \in \arg\min_{f \in \mathcal{F}_{\text{uni}}} \mathcal{L}_\lambda(f)$ such that for any linear head $w \in \mathbb{R}^k$, we have*

$$\mathbb{E}_{x \sim p_{\text{data}}}[(w^\top \hat{f}'(x) - y(x))^2] \geq 1. \tag{16}$$

We note that as an implication of the lower bound, previous works that analyze universal function approximators (Arora et al., 2019; Tosh et al., 2021; HaoChen et al., 2021) wouldn't be able to show good downstream accuracy unless the representation dimensionality is larger than $2^{d-1}$. In contrast, our theory that incorporates the inductive biases of the function class manages to show that a much lower representation dimensionality $k = s$ suffices.

We also note that this example shows a situation where Theorem 2 works but Theorem 1 doesn't, hence demonstrating how our theory derived from the eigenfunction viewpoint allows for lower representation dimensionality. There are $2^s$ model-restricted minimal clusters in the graph, each encoded by one configuration of the $s$ feature dimensions. However, all the function in $\mathcal{F}_{\text{linear}}$ that implment a cluster span in a $s$-dimensional subspace, thus we can find $s$-dimensional eigenfunctions that satisfies Assumption 6. As a result, learning $s$-dimensional representations already suffices for solving the downstream task.

## 5.2 ReLU NETWORKS

In the previous example, the downstream task is only binary classification where the label is defined by one invariant feature dimension. Here we show that when we use a ReLU network as the model architecture, the linear probing can solve more diverse downstream tasks where the label can depend on the invariant feature dimensions arbitrarily.

**Example 2.** *The natural data distribution and the data augmentation are defined in the same way as Example 1. The downstream task is a $r$-way classification problem such that the label function $y(\cdot) : \mathcal{X} \to [r]$ satisfies $y(x) = y(x')$ if $x_{1:s} = x'_{1:s}$. In other words, the label only depends on the first $s$ dimensions of the data.*

**Definition 2** (ReLU networks)**.** *Let $U \in \mathbb{R}^{k \times d}$ and $b \in \mathbb{R}^k$, we use $f_{U,b} = \sigma(Wx + b)$ to denote the ReLU network with weight $U$ and bias $b$, where $\sigma$ is the element-wise ReLU activation. We define the $k$-dimensional ReLU network function class as $\mathcal{F}_{\text{ReLU}} = \{f_{U,b} : U \in \mathbb{R}^{k \times d}, b \in \mathbb{R}^k\}$.*

We have the following theorem which shows the effectiveness of the ReLU network architecture.

**Theorem 4.** *In Example 2, suppose we set the output dimensionality $k = 2^s$ and learn a ReLU network representation map $\hat{f} = \arg\min_{f \in \mathcal{F}_{\text{ReLU}}} \mathcal{L}_\lambda(f)$ for some $\lambda > 0$. Then, we can find a linear head $W \in \mathbb{R}^{r \times k}$ such that*

$$\mathbb{E}_{x \sim p_{\text{data}}}\left[\left\| W\hat{f}(x) - e_{y(x)} \right\|_2^2\right] = 0. \tag{17}$$

*In contrast, suppose the function class is the set of universal function approximators $\mathcal{F}_{\text{uni}}$. So long as the output dimensionality is no more than $2^{d-s}$, there exists solution $\hat{f}' \in \arg\min_{f \in \mathcal{F}_{\text{uni}}} \mathcal{L}_\lambda(f)$ such that for any linear head $W \in \mathbb{R}^{r \times k}$, we have $\mathbb{E}_{x \sim p_{\text{data}}}\left[\left\| W\hat{f}'(x) - e_{y(x)} \right\|_2^2\right] \geq \frac{1}{2}$.*

## 5.3 LIPSCHITZ CONTINUOUS FUNCTIONS

In many real-world settings where a neural network is trained with weight decay, the resulting model usually has a limited weight norm which encourages the network to have a smaller Lipschitz constant. The implicit bias of the optimizers can further encourage the smoothness of the learned function. Here we provide an example showing that restricting the model class to Lipschitz continuous functions allows us to use lower dimensional representations. In particular,

we consider the following example where a large number of clusters are located close to each other despite being disconnected in the positive-pair graph. Our result shows that contrastive learning with Lipschitz continuous functions would group those clusters together, allowing for lower representation dimensionality.

**Example 3.** *Let $S_1, S_2, \cdots, S_m \subset \mathbb{R}^d$ be $m$ manifolds in $\mathbb{R}^m$, each of which may contain lots of disconnected subsets. Suppose the radius of every manifold is no larger than $\rho$, that is for any $i \in [m]$ and two data $x, x' \in S_i$, we have $\|x - x'\|_2 \leq \rho$. We also assume that different manifolds are separated by $\gamma$, that is for any $i, j \in [m]$ such that $i \neq j$, and $x \in S_i$, $x' \in S_j$, we have $\|x - x'\|_2 \geq \gamma$. The data distribution $p_{\text{data}}$ is supported on $S_1 \cup S_2 \cup \cdots \cup S_m$ and satisfies $\Pr_{x \sim p_{\text{data}}}(x \in S_i) = 1/m$ for every $i \in [m]$. A positive pair only contains data in the same $S_i$. The downstream task is a $r$-way classification problem such that the label function $y(\cdot) : \mathcal{X} \to [r]$ satisfies $y(x) = y(x')$ if $x$ and $x'$ belong to the same set $S_i$.*

We introduce the following family of Lipschitz continuous functions with parameter $\kappa$:

**Definition 3** ($\kappa$-Lipschitz continuous functions). *A function $f \in \mathbb{R}^d \to \mathbb{R}^k$ is $\kappa$-Lipschitz if $\|f(x) - f(x')\|_2 \leq \kappa \|x - x'\|_2$ for all $x, x' \in \mathbb{R}^d$. We define the $\kappa$-Lipschtiz function class $\mathcal{F}_{\text{Lip},\kappa}$ as the set of all $\kappa$-Lipschitz continuous functions in $\mathbb{R}^d \to \mathbb{R}^k$.*

We have the following theorem:

**Theorem 5.** *In Example 3, suppose $\kappa \geq \sqrt{2m}/\gamma$. Let the output dimensionality $k = m$ and learn a $\kappa$-Lipschitz continuous function $\hat{f} \in \arg\min_{\mathcal{F}_{\text{Lip},\kappa}} \mathcal{L}_\lambda(f)$ for some $\lambda > 0$. Then, we can find a linear head $W \in \mathbb{R}^{r \times k}$ such that*

$$\mathbb{E}_{x \sim p_{\text{data}}}\left[\left\|W\hat{f}(x) - e_{y(x)}\right\|_2^2\right] \leq 2rm\kappa^2\rho^2. \tag{18}$$

*On the other hand, suppose the positive-pair graph contains $\tilde{m}$ disconnected clusters, and the function class is the set of universal function approximators $\mathcal{F}_{\text{uni}}$. So long as the output dimensionality $k < \tilde{m}$, there exists solution $\hat{f}' \in \arg\min_{f \in \mathcal{F}_{\text{uni}}} \mathcal{L}_\lambda(f)$ such that for any linear head $W \in \mathbb{R}^{r \times k}$, we have $\mathbb{E}_{x \sim p_{\text{data}}}\left[\left\|W\hat{f}'(x) - e_{y(x)}\right\|_2^2\right] \geq \frac{1}{\tilde{m}}$.*

We note that a smaller $\kappa$ (hence smoother function class) decreases the RHS of equation 18 and leads to better downstream performance.

## 6 CONCLUSION

In this paper, we provide a theoretical analysis of contrastive learning that incoporates the inductive biases of the model class. We prove that contrastive learning with appropriate model architectures allows for lower representation dimensionality (hence better sample complexity), and instantiate this theory on several interesting examples. One open questions is to allow $k > m$ in our theory, which we believe requires additional assumptions on the structure of the family of the models. We note that our work only concerns the inductive biases originating from the model architecture, whereas in practice the learned representations also depend on the optimization method. Hence, another interesting future direction would be studying how the *implicit bias* introduced by the optimizer influences self-supervised learning.

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

# A  ADDITIONAL EXAMPLES

## A.1  CONVOLUTIONAL NEURAL NETWORKS

Our last example shows that convolutional neural networks can learn contrastive representation more efficiently than fully connected networks when the downstream task has a certain rotational invariance structure. We consider the following data generative model where the data contains a feature patch that determines the downstream label.

**Example 4.** *The natural data $\bar{x} \in \mathbb{R}^d$ is defined as follows: for some consecutive $s$ dimensions $\bar{x}_{t:t+s-1}$ (the informative patch), we have $\bar{x}_{t:t+s-1} \in \{-\gamma, \gamma\}^s$ where $\gamma > 1$. [1] The other $d - s$ dimensions of $\bar{x}$ (spurious dimensions) are all in $\{-1, 1\}$. Given a natural data $\bar{x}$, its augmentations are generated by first sampling $\tau \sim Uni[0, 1]$, then multiplying the spurious dimensions of $\bar{x}$ by $\tau$, while keeping the informative patch the same. The downstream task is a $r$-way classification problem such that the label function $y(\cdot) : \mathcal{X} \to [r]$ satisfies $y(x) = y(x')$ if the informative patches for $x$ and $x'$ are the same.*

We consider the following convolutional neural network model with $k$ channels.

**Definition 4** (Convolutional neural networks). *Let $U = [u_1, u_2, \cdots, u_k]^\top \in \mathbb{R}^{k \times s}$ and $b \in \mathbb{R}^k$. We use $f_{U,b}^{conv} : \mathcal{X} \to \mathbb{R}^k$ to represent the following convolutional neural network: $f_{U,b}^{conv}(x)_i = \sum_{t=1}^{d} \sigma(u_i^\top x_{t:t+s-1} + b_i)$, where $\sigma$ is ReLU activation function. We define the convolutional neural network class $\mathcal{F}_{\text{conv}} = \{f_{U,b}^{conv} : U \in \mathbb{R}^{k \times s}, b \in \mathbb{R}^k\}$.*

We have the following theorem which shows that contrastive learning with convolutional neural networks requires lower representation dimensionality than using fully-connected ReLU networks.

**Theorem 6.** *In Example 4, let output dimensionality $k = 2^s$ and learn a convolutional neural network $\hat{f} \in \arg\min_{\mathcal{F}_{\text{conv}}} \mathcal{L}_\lambda(f)$ for some $\lambda > 0$. Then, we can find a linear head $W \in \mathbb{R}^{r \times k}$ such that $\mathbb{E}_{x \sim p_{\text{data}}} \left[ \left\| W\hat{f}(x) - e_{y(x)} \right\|_2^2 \right] = 0$.*

*On the other hand, suppose the function class is the set of ReLU networks $\mathcal{F}_{ReLU}$, so long as the output dimensionality is less than $d \times 2^s$, there exists a function $\hat{f}' \in \arg\min_{f \in \mathcal{F}_{ReLU}} \mathcal{L}_\lambda(f)$ such that for any linear head $W \in \mathbb{R}^{r \times k}$, we have $\mathbb{E}_{x \sim p_{\text{data}}} \left[ \left\| W\hat{f}'(x) - e_{y(x)} \right\|_2^2 \right] \geq \frac{1}{d \cdot 2^s}$.*

# B  NUMERICAL SIMULATIONS

Recall that our assumptions intuitively state that the model architecture cannot break the data into *too many* well-separated clusters. In this section, we propose a method to empirically test how many clusters a model architecture can partition positive-pair graph of the data distribution into. Given a deep neural network and a target number of cluster $r$, ideally we aim to find a function $f$ from the model class that maps each data point to a one-hot vector in dimension $r$ which includes the cluster identity. That is, $f(x) \in \{e_1, \ldots, e_r\}$ where $e_i$ is the $i$-th natural basis in $\mathbb{R}^r$. With this constraint, we minimize the disagreement between the functions outputs of a positive-pair, that is, $\mathbb{E}_{(x,x+) \sim p_{\text{pos}}}[\|f(x) - f(x^+)\|_2^2]$, which compute the amount of inter-cluster edges. However, the one-hot vector requirement makes it challenging for optimization. Note that when the $r$ clustering has the same probability mass $1/r$, we have $\mathbb{E}[f(x)f(x)^\top] = \mathbb{I}/r$. We use this equation as the constraint of $f$ and arrive at a relaxation of the original optimization program.

$$b_r = \min \quad \mathbb{E}_{(x,x+) \sim p_{\text{pos}}}[\|f(x) - f(x^+)\|_2^2] \quad \text{s.t.} \quad \mathbb{E}[f(x)f(x)^\top] = \mathbb{I}/r \qquad (19)$$

Thus, we use $b_r$ as a surrogate for how the architecture can partition the graph into $r$ clusters, and a smaller $b_r$ means that it's easier to partition. We empirically implement equation 19 by first minimizing the contrastive loss $\mathcal{L}_\lambda(f_\theta)$ with representation dimension $k = r$ and a heavily-tuned regularization strength $\lambda$. Then, we whiten the obtained model $f_{\hat\theta}(x)$ to have exactly the covariance $\mathbb{I}/r$, that is, $\bar{f}(x) = \mathbb{E}_{x \sim p_{\text{data}}}[f_{\hat\theta}(x)f_{\hat\theta}(x)^\top]^{-\frac{1}{2}} f_{\hat\theta}(x)/\sqrt{r}$ is a valid solution for the program in equation 19. We compute $b_r = \mathbb{E}_{(x,x+) \sim p_{\text{pos}}}[\|\bar{f}(x) - \bar{f}(x^+)\|_2^2]$. We also try various choices of $\lambda$ and pick the smallest result as the final value of the estimated $b_r$.

---

[1] Here we denote $\bar{x}_{d+i} = \bar{x}_i$.

We run this test with a ResNet-18 model on CIFAR-10 and compute the $b_r$ for $r \in \{10, 100, 500\}$ list the results the table below. Here we note that $b_r$ increases from $0.127$ to $0.315$ as $r$ increases from 10 to 500, suggesting that although the network can partition the data relatively well into 10 clusters, it cannot partion the data into 500 well-separated clusters, which supports our theoretical assumptions. More details can be found in Appendix B.1.

| $r$ | 10 | 100 | 500 |
|-----|------|------|------|
| $b_r$ | 0.127 | 0.204 | 0.315 |

## B.1 Addition experimental details

We train a ResNet-18 model on CIFAR-10 training set and test the $b_r$ on the test set. We train with SGD using initial learning rate $0.01$ and decays with cosine schedule. All experiments are run for 200 epochs. We test with $r \in \{10, 100, 500\}$ and grid search using $\lambda \in \{0.1, 0.3, 1, 3, 10, 30, 100, 300, 1000\}$, the result for each configurate is listed in the table below.

| $\lambda$ | 0.1 | 0.3 | 1 | 3 | 10 | 30 | 100 | 300 | 1000 |
|-----------|-------|-------|-------|-------|-------|-------|-------|-------|-------|
| $r = 10$ | 0.445 | **0.127** | 0.134 | 0.144 | 0.151 | 0.155 | 0.215 | 0.343 | 0.901 |
| $r = 100$ | 0.887 | 0.660 | 0.408 | 0.245 | **0.204** | 0.220 | 0.254 | 0.424 | 1.579 |
| $r = 500$ | 1.031 | 0.981 | 0.710 | 0.554 | 0.427 | 0.372 | **0.315** | 0.481 | 1.231 |

## C  Proofs for Section 3

*Proof of Theorem 1.* Let $P_i := \Pr_{x \sim p_{\text{data}}}(x \in S_i)$ be the probability of $S_i$. Let $f^*$ be the function $f^*(x) = \frac{1}{\sqrt{P_{\text{id}_x}}} e_{\text{id}_x}$. From Assumption 5 and Assumption 4, we have $f^* \in \mathcal{F}$.

We first show that $f^*$ achieve small contrastive loss. For the regularizer term, we have

$$\mathbb{E}_{x \sim p_{\text{data}}} \left[ f^*(x) f^*(x)^\top \right] = \sum_{i \in [m]} P_i \cdot \frac{1}{P_i} \cdot e_{\text{id}_x} e_{\text{id}_x}^\top = \mathbb{I}. \tag{20}$$

Thus, we have $R(f^*) = 0$. For the discrepancy term, let $P_{\min} := \min_{i \in [m]} P_i$ be the probability mass of the smallest set, we have

$$\mathbb{E}_{x, x^+} \left[ \left\| f^*(x) - f^*(x^+) \right\|_2^2 \right] \leq \frac{1}{P_{\min}} \cdot \Pr_{(x, x^+) \sim p_{\text{pos}}} (\text{id}_x \neq \text{id}_{x^+}) \leq \frac{\alpha}{P_{\min}}. \tag{21}$$

Combining equation 20 and equation 21 we have

$$\mathcal{L}_\lambda(f^*) = \mathbb{E}_{x, x^+} \left[ \left\| f^*(x) - f^*(x^+) \right\|_2^2 \right] + \lambda \cdot R(f) \leq \frac{\alpha}{P_{\min}}. \tag{22}$$

Since $\hat{f} = \arg\min_{f \in F} \mathcal{L}_\lambda(f)$ is the minimizer of contrastive loss within the function class, we have

$$\mathcal{L}_\lambda(\hat{f}) \leq \mathcal{L}_\lambda(f^*) \leq \frac{\alpha}{P_{\min}}. \tag{23}$$

Define matrix

$$M := \mathbb{E}_{x \sim p_{\text{data}}} \left[ \hat{f}(x) \hat{f}(x)^\top \right]. \tag{24}$$

We have

$$\| M - \mathbb{I} \|_F^2 \leq \frac{\mathcal{L}_\lambda(\hat{f})}{\lambda} \leq \frac{\alpha}{\lambda P_{\min}}. \tag{25}$$

Since $\lambda > \frac{\alpha}{P_{\min}}$, we know that $M$ is a full rank matrix, thus we can define function

$$\tilde{f}(x) := M^{-\frac{1}{2}}\hat{f}(x). \tag{26}$$

Let

$$Q := \mathbb{E}_{x \sim p_{\text{data}}}\left[\tilde{f}(x)f^*(x)^\top\right], \tag{27}$$

and

$$\pi_f(x) := \tilde{f}(x) - Qf^*(x). \tag{28}$$

We know that

$$\mathbb{E}_{x \sim p_{\text{data}}}\left[\pi_f(x)f^*(x)^\top\right] = \mathbb{E}_{x \sim p_{\text{data}}}\left[\tilde{f}(x)f^*(x)^\top\right] - Q\mathbb{E}_{x \sim p_{\text{data}}}\left[f^*(x)f^*(x)^\top\right] = 0. \tag{29}$$

Using Assumption 3 we have:

$$\mathbb{E}_{(x,x+) \sim p_{\text{pos}}}\left[\left\|\pi_f(x) - \pi_f(x^+)\right\|_2^2\right]$$
$$\geq \sum_{i \in [m]} (P_i - \alpha) \cdot \mathbb{E}_{(x,x+) \sim p_{\text{pos}_i}}\left[\left\|\pi_f(x) - \pi_f(x^+)\right\|_2^2\right]$$
$$\geq \beta \cdot \sum_{i \in [m]} (P_i - \alpha) \cdot \mathbb{E}_{x \sim p_{\text{data}_i}, x' \sim p_{\text{data}_i}}\left[\left\|\pi_f(x) - \pi_f(x')\right\|_2^2\right]$$
$$= 2\beta \cdot \sum_{i \in [m]} (P_i - \alpha) \cdot \mathbb{E}_{x \sim p_{\text{data}_i}}\left[\left\|\pi_f(x)\right\|_2^2\right]$$
$$= 2\beta \cdot \left(1 - \frac{\alpha}{P_{\min}}\right) \cdot \mathbb{E}_{x \sim p_{\text{data}}}\left[\left\|\pi_f(x)\right\|_2^2\right]. \tag{30}$$

On the other hand, we have

$$\mathbb{E}_{(x,x+) \sim p_{\text{pos}}}\left[\left\|\pi_f(x) - \pi_f(x^+)\right\|_2^2\right]$$
$$\leq \mathbb{E}_{(x,x+) \sim p_{\text{pos}}}\left[\left\|\tilde{f}(x) - \tilde{f}(x^+)\right\|_2^2\right]$$
$$\leq \left\|M^{-1}\right\|_{\text{spec}} \cdot \mathbb{E}_{(x,x+) \sim p_{\text{pos}}}\left[\left\|\hat{f}(x) - \hat{f}(x^+)\right\|_2^2\right]$$
$$\leq \left(1 + \sqrt{\frac{\alpha}{\lambda P_{\min}}}\right) \cdot \frac{\alpha}{P_{\min}}. \tag{31}$$

Combining equation 30 and equation 31 we have

$$\mathbb{E}_{x \sim p_{\text{data}}}\left[\left\|\pi_f(x)\right\|_2^2\right] \leq \left(1 + \sqrt{\frac{\alpha}{\lambda P_{\min}}}\right) \cdot \frac{\alpha}{2\beta(P_{\min} - \alpha)}. \tag{32}$$

By Lemma 1, we know that there exists a matrix $U \in \mathbb{R}^{m \times m}$ such that

$$\mathbb{E}_{x \sim p_{\text{data}}}\left[\left\|f^*(x) - UM^{-1/2}\hat{f}(x)\right\|_2^2\right] \leq \left(1 + \sqrt{\frac{\alpha}{\lambda P_{\min}}}\right) \cdot \frac{\alpha}{2\beta(P_{\min} - \alpha)} \tag{33}$$
$$\leq \frac{\alpha}{\beta(P_{\min} - \alpha)}. \tag{34}$$

Thus, if we define matrix $W = \text{diag}\{\sqrt{P_1}, \sqrt{P_2}, \cdots, \sqrt{P_m}\}UM^{-1/2}$, then we have

$$\mathbb{E}_{x \sim p_{\text{data}}}\left[\left\|e_{\text{id}_x} - W\hat{f}(x)\right\|_2^2\right] \leq P_{\max}\mathbb{E}_{x \sim p_{\text{data}}}\left[\left\|f^*(x) - UM^{-1/2}\hat{f}(x)\right\|_2^2\right] \tag{35}$$
$$\leq P_{\max}\frac{\alpha}{\beta(P_{\min} - \alpha)}, \tag{36}$$

which finishes the proof.

$\square$

**Lemma 1.** *Suppose $f : \mathcal{X} \to \mathbb{R}^m$ and $g : \mathcal{X} \to \mathbb{R}^m$ are two functions defined on $\mathcal{X}$ such that*

$$\mathbb{E}_{x \sim p_{\text{data}}} \left[ f(x)f(x)^\top \right] = \mathbb{E}_{x \sim p_{\text{data}}} \left[ g(x)g(x)^\top \right] = \mathbb{I}. \tag{37}$$

*Define the projection of $f$ onto $g$'s orthogonal subspace as:*

$$\pi_f(x) = f(x) - \mathbb{E}_{x' \sim p_{\text{data}}} \left[ f(x)g(x)^\top \right] g(x). \tag{38}$$

*Then, there exists matrix $U \in \mathbb{R}^{m \times m}$ such that*

$$\mathbb{E}_{x \sim p_{\text{data}}} \left[ \| g(x) - Uf(x) \|_2^2 \right] = \mathbb{E}_{x \sim p_{\text{data}}} \left[ \| \pi_f(x) \|_2^2 \right]. \tag{39}$$

*Proof of Lemma 1.* Let matrix

$$U = \mathbb{E}_{x' \sim p_{\text{data}}} \left[ g(x)f(x)^\top \right]. \tag{40}$$

We have

$$\mathbb{E}_{x \sim p_{\text{data}}} \left[ \| g(x) - Uf(x) \|_2^2 \right] \tag{41}$$

$$= \mathbb{E}_{x \sim p_{\text{data}}} \left[ \| g(x) \|_2^2 \right] - 2\mathbb{E}_{x \sim p_{\text{data}}} \left[ g(x)^\top Uf(x) \right] + \mathbb{E}_{x \sim p_{\text{data}}} \left[ f(x)^\top U^\top Uf(x) \right] \tag{42}$$

$$= m - 2 \| U \|_F^2 + \| U \|_F^2 \tag{43}$$

$$= m - \| U \|_F^2. \tag{44}$$

On the other hand, we have

$$\mathbb{E}_{x \sim p_{\text{data}}} \left[ \| \pi_f(x) \|_2^2 \right] \tag{45}$$

$$= \mathbb{E}_{x \sim p_{\text{data}}} \left[ \| f(x) - U^\top g(x) \|_2^2 \right] \tag{46}$$

$$= \mathbb{E}_{x \sim p_{\text{data}}} \left[ \| f(x) \|_2^2 \right] - 2\mathbb{E}_{x \sim p_{\text{data}}} \left[ f(x)^\top U^\top g(x) \right] + \mathbb{E}_{x \sim p_{\text{data}}} \left[ g(x)^\top UU^\top g(x) \right] \tag{47}$$

$$= m - \| U \|_F^2. \tag{48}$$

Thus, we have

$$\mathbb{E}_{x \sim p_{\text{data}}} \left[ \| g(x) - Uf(x) \|_2^2 \right] = \mathbb{E}_{x \sim p_{\text{data}}} \left[ \| \pi_f(x) \|_2^2 \right], \tag{49}$$

which finishes the proof. $\qquad \square$

## D  PROOFS FOR SECTION 4

*Proof of Proposition 1.* Define function

$$\tilde{g}(x) = \sqrt{p_{\text{data}}(x)} g(x). \tag{50}$$

Define the symmetric Laplacian operator

$$\widetilde{\mathbb{L}}(\tilde{g})(x) = \tilde{g}(x) - \int \frac{p_{\text{pos}}(x, x')}{\sqrt{p_{\text{data}}(x)}\sqrt{p_{\text{data}}(x')}} \tilde{g}(x') dx'. \tag{51}$$

It can be verified that

$$\int_x \tilde{g}(x) \widetilde{\mathbb{L}}(\tilde{g})(x) = 0. \tag{52}$$

Notice that the operator $\widetilde{\mathbb{L}}$ is PSD, we have that

$$\int_x (\widetilde{\mathbb{L}}(\tilde{g})(x))^2 = 0, \tag{53}$$

which is equivalent to

$$\mathbb{E}_{x \sim p_{\text{data}}} \left[ (\mathbb{L}(\tilde{g})(x))^2 \right] = 0, \tag{54}$$

hence finishes the proof. $\qquad \square$

*Proof of Theorem 2.* Notice that $\mathcal{L}_\lambda(f_{\text{eig}}) \leq \phi$, we know that $\mathcal{L}_\lambda(\hat{f}) \leq \phi$, so

$$\left\| \mathbb{E}_{x \sim p_{\text{data}}} \left[ \hat{f}(x)\hat{f}(x)^\top \right] - \mathbb{I} \right\|_F^2 \leq \frac{\phi}{\lambda}. \tag{55}$$

In Assumption 6, set $f = \hat{f}$ and sum over $i = 1, 2, \cdots, k$, we have that for some matrix $\tilde{W}$,

$$\mathbb{E}_{x \sim p_{\text{data}}} \left[ \left\| \tilde{W} f_{\text{eig}}(x) - \hat{f}(x) \right\|_2^2 \right] \leq k\epsilon. \tag{56}$$

Let matrix $Q := \mathbb{E}_{x \sim p_{\text{data}}}[\hat{f}(x)\hat{f}(x)^\top]$, we have that

$$\mathbb{E}_{x \sim p_{\text{data}}} \left[ \left\| Q^{-1/2}\hat{f}(x) - \hat{f}(x) \right\|_2^2 \right] \leq \frac{2\phi}{\lambda} \cdot \mathbb{E}_{x \sim p_{\text{data}}} \left[ \left\| \hat{f}(x) \right\|_2^2 \right] \leq \frac{2\phi}{\lambda} k \left( 1 + \sqrt{\frac{\phi}{\lambda}} \right). \tag{57}$$

Thus,

$$\mathbb{E}_{x \sim p_{\text{data}}} \left[ \left\| \tilde{W} f_{\text{eig}}(x) - Q^{-1/2}\hat{f}(x) \right\|_2^2 \right] \leq 2k\epsilon + \frac{4\phi}{\lambda} k \left( 1 + \sqrt{\frac{\phi}{\lambda}} \right). \tag{58}$$

Define matrix

$$M := \mathbb{E}_{x \sim p_{\text{data}}} \left[ f_{\text{eig}}(x) Q^{-1/2}\hat{f}(x)^\top \right] \tag{59}$$

Using Lemma 1 and equation 58 we have

$$\mathbb{E}_{x \sim p_{\text{data}}} \left[ \left\| f_{\text{eig}}(x) - MQ^{-1/2}\hat{f}(x) \right\|_2^2 \right] \leq 2k\epsilon + \frac{4\phi}{\lambda} k \left( 1 + \sqrt{\frac{\phi}{\lambda}} \right) \leq 2k\epsilon + \frac{8\phi}{\lambda} k. \tag{60}$$

Thus, using Assumption 7, we have

$$\mathbb{E}_{x \sim p_{\text{data}}} \left[ \left\| W^* MQ^{-1/2}\hat{f}(x) - e_{y(x)} \right\|_2^2 \right] \tag{61}$$

$$\leq 2\mathbb{E}_{x \sim p_{\text{data}}} \left[ \left\| W^* f_{\text{eig}}(x) - e_{y(x)} \right\|_2^2 \right] + 2\mathbb{E}_{x \sim p_{\text{data}}} \left[ \left\| W^* MQ^{-1/2}\hat{f}(x) - W^* f_{\text{eig}}(x) \right\|_2^2 \right] \tag{62}$$

$$\leq 2\zeta + 4B^2 k\epsilon + \frac{16\phi}{\lambda} B^2 k. \tag{63}$$

$\square$

# E    PROOFS FOR SECTION 5

## E.1    PROOF FOR EXAMPLE 1

*Proof of Theorem 3.* Define $\hat{U} = [e_1, e_2, \cdots, e_s]^\top \in \mathbb{R}^{s \times d}$. We can verify that

$$\mathbb{E}_{(x,x^+) \sim p_{\text{pos}}} \left[ \left\| f_{\hat{U}}(x) - f_{\hat{U}}(x^+) \right\|_2^2 \right] = 0 \tag{64}$$

and

$$\mathbb{E}_{x \sim p_{\text{data}}} \left[ f_{\hat{U}}(x) f_{\hat{U}}(x)^\top \right] = \mathbb{I}. \tag{65}$$

Thus, we can view $f_{\hat{U}}$ as the $f_{\text{eig}}$ in Section 4.

Let $U \in \mathbb{R}^{k \times d}$ and $i \in [k]$ such that

$$\mathbb{E}_{(x,x^+) \sim p_{\text{pos}}}[(f_U(x)_i - f_U(x^+)_i)^2] = 0. \tag{66}$$

Notice that $x$ and $x^+$ only differs on the $s + 1$-th to $d$-th dimensions, we know that $U_i$ is 0 on the $s + 1$-th to $d$-th dimensions. Thus, we have that $U_i$ is in the span of $e_1, e_2, \cdots, e_s$, and as a result Assumption 6 holds with $\epsilon = 0$.

Since the downstream task's label is equal to $x_i$ for $i \in [s]$, we can set $W^* = e_i^\top$ and we would have

$$\mathbb{E}_{x \sim p_{\text{data}}} \left[ (W^* f_{\hat{U}}(x) - \vec{y}(x))^2 \right] = 0. \tag{67}$$

Hence Assumption 7 holds with $\alpha = 0$ and $B = 1$.

Applying Theorem 2 finishes the proof for the linear function class case.

For the case of universal function approximators, without loss of generality we assume the downstream task's label only depends on the first dimension of $x$, i.e., $y(x) = \text{sgn}(x_1)$. When $k \leq 2^{d-1}$, we can construct a function $f : \mathcal{X} \to \mathbb{R}^k$ such that for every diemnsion $j \in [k]$, we have $f(x)_j = \sqrt{k}$ when $x_{2:d}$ viewed as a binary number equals to $j$, otherwise $f(x)_j = 0$. It can be verified that $\mathcal{L}_\lambda(f) = 0$ hence $f$ is a minimizer of the contrastive loss. However, $f(x)$ is agnostic to the first dimension of $x$, hence the downstream error is at least 1. $\qquad\square$

### E.2 PROOF FOR EXAMPLE 2

*Proof of Theorem 4.* For any vector $h \in \{-1, 1\}^s$, we define function $\text{bin}(h) \in \{0, 1, \cdots, 2^s - 1\}$ be the function that maps $h$ to the corresponding number when viewing $\frac{1}{2}(h + 1)$ as binary. Since $\text{bin}(\cdot)$ is a one-to-one mapping, we can define $U \in \{k \times d\}$ such that the $i$-th row of $U$ satisfies: the first $s$ dimensions equal $\sqrt{k} \cdot \text{bin}^{-1}(i - 1)$, and the rest $d - s$ dimensions are 0. Let bias vector $b \in \mathbb{R}^k$ such that every dimension is $-\sqrt{k} \cdot (r - 1)$. We have $f_{U,b}(x) = \sqrt{k} \cdot e_{\text{bin}(x_{1:s})+1} \in \mathbb{R}^k$.

Since $\mathbb{E}_{x \sim p_{\text{data}}}[f_{U,b}(x) f_{U,b}(x)^\top] = \mathbb{I}$ and $\mathbb{E}_{(x,x^+) \sim p_{\text{pos}}}[\|f_{U,b}(x) - f_{U,b}(x^+)\|_2^2] = 0$, we can view $f_{U,b}$ as the $f_{\text{eig}}$ in Section 4. Assumption 7 naturally hold wihth $B = 1$.

For Assumption 6, consider a function $f_{U',b'} \in \mathcal{F}_{\text{ReLU}}$ and index $i \in [k]$ such that $\mathbb{E}_{(x,x^+) \sim p_{\text{pos}}} \left[ (f_{U',b'}(x)_i - f_{U',b'}(x^+)_i)^2 \right] = 0$. Suppose there exist $\bar{x} \neq \bar{x}'$ and their augmentations $x, x'$ such that $f_{U',b'}(x)_i > f_{U',b'}(x')_i$. Then, there must be $(U_i)_{r+1:d} \neq 0$ and $\sigma(U_i^\top x) > 0$. This suggests that there must exist another $\tilde{x}$ which is also an augmentation of $\bar{x}$ but $\sigma(U_i^\top x) \neq \sigma(U_i^\top \tilde{x})$. Hence, we have

$$\mathbb{E}_{(x,x^+) \sim p_{\text{pos}}} \left[ (f_{U',b'}(x)_i - f_{U',b'}(x^+)_i)^2 \right] > 0, \tag{68}$$

leading to contradiction. Hence, we know that $f_{U',b'}(x)_i = f_{U',b'}(x')_i$, so $(f_{U',b'})_i$ can only be a function of $x_{1:s}$. Therefore, there exists a vector $w \in \mathbb{R}^k$ such that $f_{U',b'}(x)_i = w^\top f_{U,b}(x)$, which means Assumption 6 holds with $\epsilon = 0$. Applying Theorem 2 finishes the proof for equation 17.

The result about universal function approximators follows the same proof as for Theorem 3 execpt for constructing the function using the last $(d - s)$ dimensions rather than the last $(d - 1)$ dimensions. $\qquad\square$

### E.3 PROOF FOR EXAMPLE 3

*Proof of Theorem 5.* Let $\text{id}_x$ be the index such that $x \in S_{\text{id}_x}$, and define function $f_{\text{eig}}(x) = \sqrt{m} \cdot e_{\text{id}_x}$. It can be verified that $f_{\text{eig}}$ satisfies equation 9 and equation 10. For $f \in \mathcal{F}_{\text{Lip},\kappa}$ and $i \in [m]$, define $g(x) = f(x)_i$. Suppose $\mathbb{E}_{x \sim p_{\text{data}}}[g(x)^2] = 1$, we can chooose $m$ data $x_1, x_2, \cdots, x_m$ such that $x_i \in S_i$ and $\frac{1}{m} \sum_{i \in [m]} g(x_i)^2 \leq 1$. Define vector $\tilde{w} \in \mathbb{R}^m$ such that $\tilde{w}_i = \frac{1}{\sqrt{m}} \cdot g(x_i)$. We have

$$\mathbb{E}_{x \sim p_{\text{data}}} \left[ (\tilde{w}^\top f_{\text{eig}}(x) - g(x))^2 \right] = \frac{1}{m} \sum_{i \in [m]} \mathbb{E}_{x \sim S_i} \left[ (g(x_i) - g(x))^2 \right] \tag{69}$$

$$\leq \frac{1}{m} \sum_{i \in [m]} \kappa^2 \rho^2 = \kappa^2 \rho^2. \tag{70}$$

Thus, $f_{\text{eig}}$ satisfies Assumption 6 with $\epsilon = \kappa^2 \rho^2$.

Since the data in the same $S_i$ have the same downstream label, we know that Assumption 7 holds with $B = \sqrt{r}$ and $\alpha = 0$. Thus, applying Theorem 2 finishes the proof for the upper bound.

For the lower bound, Let set $\tilde{S}$ be the set among those $\tilde{m}$ clusters that has the largest size. When $k < \tilde{m}$, we can construct a function that maps all data in $\tilde{S}$ to 0, hence the final error would be at least $\frac{1}{\tilde{m}}$. $\qquad\square$

### E.4 PROOF FOR EXAMPLE 4

*Proof of Theorem 6.* For any vector $h \in \{-1, 1\}^s$, we define function $\text{bin}(h) \in \{0, 1, \cdots, 2^s - 1\}$ be the function that maps $h$ to the corresponding number when viewing $\frac{1}{2}(h + 1)$ as binary. Since $\text{bin}(\cdot)$ is a one-to-one mapping, we can define $U \in \{k \times s\}$ such that the $i$-th row of $U$ equal to $\frac{\sqrt{k}}{\gamma-1} \cdot \text{bin}^{-1}(i - 1)$. Let bias vector $b \in \mathbb{R}^k$ be such that every dimension is $\frac{\sqrt{k}}{\gamma-1}(-s - (\gamma - 1)(s - 1))$. We have $f_{U,b}^{\text{conv}}(x) = \sqrt{k} \cdot e_{\text{bin}(x_{t:t+s-1})+1} \in \mathbb{R}^k$, where $t$ is the starting position of the informative patch in $x$. It can be verified that $\mathbb{E}_{x \sim p_{\text{data}}}[f_{U,b}^{\text{conv}}(x) f_{U,b}^{\text{conv}}(x)^\top] = \mathbb{I}$ and $\mathbb{E}_{(x,x^+) \sim p_{\text{pos}}}[\left\| f_{U,b}^{\text{conv}}(x) - f_{U,b}^{\text{conv}}(x^+) \right\|_2^2] = 0$. Also, Assumption 7 holds with $B = 1$ when viewing $f_{\text{eig}} = f_{U,b}^{\text{conv}}$.

Suppose some function $f_{\hat{U},\hat{b}}^{\text{conv}}$ and dimension $i \in [k]$ satisfies

$$\mathbb{E}_{(x,x^+) \sim p_{\text{pos}}} \left[ \left( f_{\hat{U},\hat{b}}^{\text{conv}}(x)_i - f_{\hat{U},\hat{b}}^{\text{conv}}(x^+)_i \right)^2 \right] = 0. \tag{71}$$

Then, we know that for any $x \in p_{\text{data}}$, suppose we define $\tilde{x}$ as the vector that replaces spurious dimensions of $x$ with 0. Notice that $\tilde{x}$ is in the support of $x$'s augmentations, and the model is continuous, we know $f_{\hat{U},\hat{b}}^{\text{conv}}(x)_i = f_{\hat{U},\hat{b}}^{\text{conv}}(\tilde{x})_i$ Further notice that for any two data $x, x'$ with the same informative patch (location might be different) and corresponding $\tilde{x}, \tilde{x}'$, there must be $f_{\hat{U},\hat{b}}^{\text{conv}}(\tilde{x})_i = f_{\hat{U},\hat{b}}^{\text{conv}}(\tilde{x}')_i$ due to the structure of the convolutional neural networks. Thus, We have $f_{\hat{U},\hat{b}}^{\text{conv}}(x)_i = f_{\hat{U},\hat{b}}^{\text{conv}}(x')_i$. This suggests that the funciton $f_{\hat{U},\hat{b}}^{\text{conv}}(\cdot)_i$ is in the span of $f_{U,b}^{\text{conv}}$, hence finishes the proof for the upper bound.

For the lower bound, we note that due to the lack of invariance to informative patch location, we can construct a network with $d \cdot 2^s$-dimensional output that satisfies equation 9 and equation 10. When then output dimension is less than $d \cdot 2^s$, there would exist a minimizer of the contrastive loss that merges two of these $d \cdot 2^s$ clusters. If these two clusters have different downstream label, there would be at least $\frac{1}{d \cdot 2^s}$ loss incurred due to the data being mapped to the same feature, hence finishes the proof for the lower bound. $\qquad \square$

