# OpenReview forum: "A theoretical study of inductive biases in contrastive learning"
_ICLR.cc/2023/Conference — ICLR 2023 poster_

### Official Review · Reviewer_HMnY · 2022-10-24

**Confidence:** 4
**Correctness:** 3
**Technical Novelty And Significance:** 3
**Empirical Novelty And Significance:** Not applicable
**Recommendation:** 6

**Clarity, Quality, Novelty And Reproducibility:**

Clarity: The paper has good clarification and motivation.

Quality: I checked most proofs in the appendix. The statements look good to me.

Novelty: The paper has its novelty as far as I know.

Reproducibility: I believe the experiments part can be reproduced.


**Strength And Weaknesses:**

Strength:
1. As far as I know, this is the first work to study inductive bias originating from the model class in contrastive learning. The paper is based on some reasonable assumptions and gets a non-trivial conclusion. In my opinion, Assumption 2 is the most critical one. The function class cannot be too powerful so that we can guarantee the generalization. The authors also introduce novel tools, eigenfunction methods, in analyzing contrastive learning.
2. The paper has a clear motivation and a good structure.

Weakness and Questions:
1.  I am not familiar with Eignfunction methods. I have some questions for Section 4. What is the relationship between $k$ orthogonal eigenfunctions and equations (9) and (10)? Are equations (9) and (10) additional assumptions here or can they be induced from $k$ orthogonal eigenfunctions? What is the connection between Assumption 6 and Assumption 2 and 3? I cannot get full insights into how is Therom 2 generalized from Theorem 1.
2. One concern I have is that in practice the model has more than millions of parameters, e.g. vision transformer. Then, how to guarantee assumption 2 in a real empirical scenario. Note that Assumption 2 considers the whole function space. If assumption 2 is not guaranteed, how to explain the success in MoCo v3 [3]?
3. There are some subtle weaknesses in the problem settings. The paper studies generalized spectral contrastive loss rather than the InfoNCE family loss. The authors also did not consider the sample complexity in the pre-training like [1,2].
4. The experiments part is weak.  It is acceptable considering it is a theoretical paper. The $b_r$ is not a clear metric for reader to understand the partition effect, e.g. for $r=500$, how bad is $b_r=0.315$. More explanation is needed here.

Typos:
1. In Figure 2, there are no orange points.
2. In page 16, “wiht” -> “with”.

[1] Arora, Sanjeev, et al. "A theoretical analysis of contrastive unsupervised representation learning." arXiv preprint arXiv:1902.09229 (2019).

[2] HaoChen, Jeff Z., et al. "Provable guarantees for self-supervised deep learning with spectral contrastive loss." Advances in Neural Information Processing Systems 34 (2021): 5000-5011.

[3] Chen, Xinlei, Saining Xie, and Kaiming He. "An empirical study of training self-supervised vision transformers." Proceedings of the IEEE/CVF International Conference on Computer Vision. 2021.


**Summary Of The Paper:**

The paper studies inductive bias from the model class in contrastive learning. The paper proposes some mild and practical assumptions and studies the optimal solution of the spectral contrastive loss from the eigenfunction perspective. The paper argues that when the model has limited capacity, contrastive representations would recover certain special clustering structures (depending on model architecture) but ignore other clustering structures in the data distribution. Moreover, the paper also studies different model classes including linear functions, ReLU networks, Lipschitz continuous functions, and Convolutional neural networks.

**Summary Of The Review:**

Although there are some questions I mentioned in the Weakness part that blocked me, the paper has its novelty and I tend to accept it.

---

> ### Author Response · Authors · 2022-11-12
> **Response to Reviewer HMnY**
>
> We thank the reviewer for their comments and their tendency to accept our work. The reviewer appreciates that our work “is based on some reasonable assumptions” and “gets a non-trivial conclusion”, that we “introduce novel tools”, and that the paper “has a clear motivation and a good structure”.
>
> We want to emphasize that the core contribution of this work is to provide the **first theoretical analyses** of how inductive biases originating from the model class lead to better contrastive learned representations, shedding light on a recently observed empirical mystery. As we discussed in the general response, the numeral experiments are only complementary.
>
> We address the comments of the reviewers below.
>
> > Explanation of equations (9) and (10)
>
> Equations (9) and (10) are rigorous assumptions, whereas “k orthogonal eigenfunctions” is mentioned in our paper only as an explanation of these assumptions. Intuitively, equations (9) and (10) together imply that f_eig is roughly (but not necessarily exactly) formed by k orthogonal eigenfunctions.
>
> > How Theorem 2 generalizes Theorem 1
>
> Recall that Theorem 1 assumes that the positive-pair graph is composed of several minimal implementable clusters, where the identity function of each cluster can be implemented by some function in the function class. Since the clusters have a small connection to each other, such identity functions would also be an achievable eigenfunction, which is the assumption in Theorem 2.
>
> > How to guarantee assumption 2 in a real empirical scenario.
>
> We agree that our work is mainly theoretical, and it can be challenging to show that empirically assumption 2 is satisfied, especially when the neural network model is large. Our experiments made a first step towards showing that Assumption 2 holds for reasonably larger models like ResNet 50. We believe that for larger models Assumption 2 can still hold if we also take into consideration the implicit regularization of the optimizer, which might require additional theoretical tools to analyze and remains an open question.
>
> > Generalized spectral contrastive loss vs InfoNCE family loss.
>
> Although InfoNCE is more widely used in practice, the theoretical analysis of it still remains a challenging open question. We study generalized spectral contrastive loss because it’s more amenable to rigorous end-to-end theoretical analysis, which is a common practice in the literature [HaoChen et al. 2021].
>
> > More explanation of the empirical partition metric b_r
>
> The value b_r intuitively captures the portion of edges that are crossing different clusters, and is always in the range of [0, 1]. When b_r=0, there’s no edge between clusters in the positive-pair graph, whereas b_r=1 means that all the edges are between clusters. b_10 \approx 0.1 means that roughly 10% of edges are across the 10-way partition clusters, which intuitively corresponds to a 10% error rate after supervised linear probing.
>
> Given that it’s challenging to empirically test the assumptions in our theorem, we propose this b_r metric as our best effort to achieve that. We note that the quick increase from b_10 to b_500 suggests that there’s a fundamental challenge for the model to partition the data into many clusters, but it’s relatively easy for it to partition the graph into only a few clusters. This observation supports our assumptions in sec 3.
>
> Reference:
>
> [HaoChen et al. 2021] Provable Guarantees for Self-Supervised Deep Learning with Spectral Contrastive Loss. NeurIPS 2021.

---

> ### Comment · Reviewer_HMnY · 2022-11-17
> **Appreciate authors' response**
>
> After checking the rebuttal and other reviewers’ comments, I tend to keep my score. The paper will be much stronger if the author can fix any problem below (1) k>m case (2) implicit regularization of the optimization to guarantee the assumption (3) some analysis based on InfoNCE family loss (4) some direct empirical verification or support to the theory.

---

### Official Review · Reviewer_XWTN · 2022-10-25

**Confidence:** 3
**Correctness:** 4
**Technical Novelty And Significance:** 3
**Empirical Novelty And Significance:** 2
**Recommendation:** 6

**Clarity, Quality, Novelty And Reproducibility:**

The work is clear and high-quality. I am not an expert in contrastive learning and cannot judge the novelty---I also skimmed the proofs for correctness but cannot attest to how theoretically interesting they are.

**Strength And Weaknesses:**

Strengths:

- The paper is well-written and the authors provide intuition around each provided result, and motivate the problem well.
- The authors present the example from Saunshi et al (2021) and show how their results shed light on the actual behavior of contrastive learning in that setting.

Weaknesses:

- The experimental section is a bit confusing and I don't quite see the connection between the theory and the experimental results. In particular, the experiment shows that as the number of clusters gets higher, the loss of a contrastive classifier also gets higher. My issue with this experiment is that I do not see what the alternative is to the results shown in the table in page 9---clearly as r -> infinity the loss must increase, and for r = 1 the objective is trivial, so I'm not sure what I should take away from the loss increasing (the numbers themselves are not interpretable to me).
- A much more compelling experiment would be to show that the bounds proved in the previous sections are actually predictive of model behavior in some non-obvious way (e.g., by implicitly varying P_min and P_max, or by varying alpha and beta) and showing that Theorem 1 is predictive in this setting (at least in terms of trend)

**Summary Of The Paper:**

This paper provides a theoretical analysis of contrastive learning in the setting where the hypothesis class has significantly smaller dimension than the data distribution (as measured by the number of clusters). They instantiate their framework on a few example settings and show that their results provide theoretical guarantees where previous approaches do not.

**Summary Of The Review:**

This paper presents a theoretical analysis of inductive biases in contrastive learning. Their main result is that when the function class being learned over is restricted, one needs a representation size that is far smaller than the number of clusters in the data distribution. The paper is clearly written, the results seem correct, and also address issues raised by prior work.

---

> ### Author Response · Authors · 2022-11-12
> **Response to Reviewer XWTN**
>
> We thank the reviewer for their thoughtful comments, for saying that we “motivate the problem well” and “address issues raised by prior work”, and that our work “is clear and high-quality”.
>
> Our core contribution is to provide **theoretical analyses** of how inductive biases originating from the model class lead to better contrastive learned representations. As far as we know, we provide the **first** theoretical analysis of self-supervised learning that incorporates the effect of inductive biases, shedding light on a recently observed empirical mystery [Saunshi et al. 2022].
>
> We acknowledge the reviewer has concerns about experiments. We want to emphasize that our main contribution is the theory, and experiments are only complementary.
>
> More discussion on the experiments:
> - Given that it’s challenging to empirically test the assumptions in our theorem, we provide experiments (in particular, the optimization-then-whitening algorithm) that are our best effort to achieve that. The quick increase from b_10 to b_500 suggests that there’s a fundamental challenge for the model to partition the data into many clusters, but it’s relatively easy for it to partition the graph into only a few clusters. This observation supports our assumptions in sec 3.
> - The value b_r intuitively captures the portion of edges that are crossing different clusters, and is always in the range of [0, 1]. When b_r=0, there’s no edge between clusters in the positive-pair graph, whereas b_r=1 means that all the edges are between clusters. b_10 \approx 0.1 means that roughly 10% of edges are across the 10-way partition clusters, which intuitively corresponds to a 10% error rate after supervised linear probing.
>
> Reference:
>
> [Saunshi et al. 2022] Understanding contrastive learning requires incorporating inductive biases. ICML 2022.

---

### Official Review · Reviewer_1bvx · 2022-10-26

**Confidence:** 4
**Correctness:** 3
**Technical Novelty And Significance:** 3
**Empirical Novelty And Significance:** 2
**Recommendation:** 6

**Clarity, Quality, Novelty And Reproducibility:**

The results are novel and reproducible from the proofs, however as discussed before, the paper requires a lot of work on the writing side to be accessible to the ICLR audience. Here are some particular suggestions:
- Fix Figure 1
- Explain HaoChen et al. 2021, particularly the positive-pair graph. Maybe add a figure.
- Further emphasize the role of dimensionality here, and the fact that the improvement is in terms of this
- Since the $m$-way partition is defined independent of $\mathcal{F}$, and tied to that using Assumption 1 & 2, it took me a bit of back and forth to understand. Would be great to abstract the definition separately from the assumptions, and then talk about the assumptions.
- Add more discussion regarding the assumptions, as to why they are necessary, especially Assumption 4, which seems redundant given that we use a linear map on top to predict anyway. Furthermore, the setting could be defined first (which includes assumption 1 & 5 which are only assumptions on the data generating process), and then the assumptions on the function class.
- Would be good to add a comparison to HaoChen et al. 2021, and whether you get their results assuming $\mathcal{F}$ is the class of all functions.
- More discussion on why the condition in the eigenfunction section compares to the assumption in the previous section, how the quantities relate
- Typos:
    - Wrong paper cited in the abstract
    - \cite → \citep in appropriate places

**Strength And Weaknesses:**

**Strengths**:
- The main result in the paper improves the current theoretical understanding of self-supervised learning and formalizes the example presented in Saunshi et al. (2020).
- The eigenfunction perspective is a nice generalization and probably has connections directly to spectral clustering/kernel PCA.
- The authors do a good job of including several examples to instantiate the theorem.

**Weaknesses**:
- The main weakness of the paper is writing. The paper is written for a narrow set of the audience who is very familiar with the existing works of HaoChen et al. 2021, 2022 and Saunshi et al. 2022. Very little effort has been put to make the core ideas accessible to a broader audience. Furthermore, figure 1 is very hard to understand given that it has no “orange” points. Considering this is the main figure used to motivate the underlying study, this already makes it very hard to follow the paper.
- The experimental evaluation is not very convincing. Assumption 2 depends on all $f \in \mathcal{F}$ but the evaluation is based on a specific value of the learned $f$ from the contrastive solution (which encodes the inductive biases of the algorithm as well). It is hard to understand what these numbers mean and whether the trend is enough.

**Summary Of The Paper:**

Prior work by Saunshi et al. 2022 had highlighted the insufficiency of previous theoretical analyses that are function-class agnostic to completely explain the success of self-supervised learning. This paper provides theoretical guarantees for self-supervised learning that do incorporate the function class building on the framework of HaoChen et al 2021.

In particular, the paper presents a notion of *minimal implementable clusters* for a function class $\mathcal{F}$, that is, the size of the smallest disjoint partition of the input space such that the positive pairs lie with high probability in the same cluster and the clusters within each partition are non-sparse under the geometry enforced by the underlying function class $\mathcal{F}$. At a high level, this notion intends to captures the ability of the function class $\mathcal{F}$ to break the underlying positive-pair graph (vertices = augmentations of images, edges = if two augmentations can be generated from the same image) into too many clusters. The main theorem then says that the embedding dimension needed to guarantee good downstream performance is dependent on the minimal implementable clusters which may be exponentially smaller than the total number of clusters independent of the the function class.

**Summary Of The Review:**

Overall, I think the paper makes progress towards understanding the role of function class for contrastive learning and formalizing the observations in Saunshi et al. 2022. Given that the prior work did already propose a setting where the role of function class was highlighted, this paper’s primary contribution is a more refined theoretical treatment of this observation, and not the observation itself. This diminishes the contribution of the work slightly, however I still think it would be interesting to the ICLR audience. My major concern is writing and the experimental section, which is reflected in my comments above. Therefore, currently I think the paper is not ready for acceptance. I will be happy to increase my score if these are improved.

Post rebuttal: 5-> 6

---

> ### Author Response · Authors · 2022-11-12
> **Response to Reviewer 1bvx**
>
> We thank the reviewer for their insightful suggestions, for appreciating that our work “improves the current theoretical understanding of self-supervised learning”, that the “eigenfunction perspective is a nice generalization”, and that we “do a good job of including several examples to instantiate the theorem”.
>
> The major concern of the reviewer is the writing. We’ve updated our paper according to the reviewer’s suggestions, and we believe this makes our writing much more clear and our paper much stronger. Concretely, we made the following changes (the changed text is marked with orange color in our new revision):
> 1. Updated Figure 1 to include the orange points
> 2. Added more explanation of the positive-pair graph in Sec 3. Also added a figure demonstrating the positive-pair graph.
> 3. Added one paragraph in sec 3 discussing the main contribution of this work compared with the prior work, i.e., the reduction of feature dimensionality.
> 4. Improved the structure of sec 3 to make the definition of partitions/downstream closer to each other, and move assumptions about the function classes later.
> 5. Added more discussion regarding the “closure under scaling” assumption.
> 6. Added a paragraph discussing the relationship between our result and [HaoChen et al 2021].
> 7. Added a paragraph discussing the relationship between Theorem 2 and Theorem 1.
>
> We want to emphasize that the core contribution of this work is to provide the **first theoretical analyses** of how inductive biases originating from the model class lead to better contrastive learned representations, shedding light on a recently observed empirical mystery. As we discussed in the general response, the numeral experiments are only complementary.
>
> We thank the reviewer for saying that “I will be happy to increase my score if these are improved”. Since we’ve addressed the main concern raised by the reviewer and updated the paper accordingly, we would appreciate it if the reviewer can increase their score.
>
> Our responses to more specific comments are below.
>
> > The paper is written for a narrow set of the audience who is very familiar with the existing works.
>
> We thank the reviewer for pointing out this writing issue of our paper. Indeed, our work is built on top of the prior works [HaoChen et al. 2021] and [Saunshi et al. 2022], and we shortened our discussion of their works (in the introduction part) due to the page limitation. We’ve added more related discussions in our revision as the reviewer suggested!
>
> > The experimental evaluation is not very convincing.
>
> We agree that it’s challenging to empirically test our theorem, and our experiments (in particular, the optimization-then-whitening algorithm) is our best effort to achieve that. We’d like to emphasize that our main contribution is the theory, whereas the experiments are just supplementary.
>
> References:
>
> [Saunshi et al. 2022] Understanding contrastive learning requires incorporating inductive biases. ICML 2022.
>
> [HaoChen et al. 2021] Provable Guarantees for Self-Supervised Deep Learning with Spectral Contrastive Loss. NeurIPS 2021.

---

> > ### Comment · Reviewer_1bvx · 2022-11-15
> > **Response to Rebuttal**
> >
> > Thank you for giving a detailed response to my comments/concerns. I
> > - The new exposition is definitely better and easier to follow. Glad to see that most of my suggestions have been incorporated.
> > - Figure 1 is still confusing to understand. Might be worthwhile to draw the linear function in the left part. Also, clarifying implementable being something that matches the exact value is important. For instance if this was a linear threshold (which makes more sense for binary output), both would be implementable.
> > - Could you explain why you need "closure under scaling" assumption? Where does the proof break without it?
> > - __mainly because when $\mathcal{F}$ has limited capacity, a higher dimensional feature may contain a lot of “wrong features” while omitting the “right features"__  Could you explain this? This seems to suggest that the result breaks down if $k\ne m$. There should be some flexibility here.
> > - Do the authors have any ideas on how to get a more rigorous experimental evaluation? If not, the authors should tone down that section and its claims.
> >
> > I would like the authors to address these before I increase my score.

---

> > > ### Author Response · Authors · 2022-11-17
> > > **Further Response**
> > >
> > > We thank the reviewer for their additional comments.
> > >
> > > > Figure 1 is still confusing to understand. Might be worthwhile to draw the linear function in the left part. Also, clarifying implementable being something that matches the exact value is important. For instance, if this was a linear threshold (which makes more sense for binary output), both would be implementable.
> > >
> > > Thanks for these great suggestions! We’ve updated the figure and its captions, emphasizing that implementating means exactly matching the values.
> > >
> > > > Could you explain why you need "closure under scaling" assumption? Where does the proof break without it?
> > >
> > > Our assumption 4 guarantees that $e_{id_x}$ is implementable. However, for the proof to work, we need $\frac{1}{\sqrt{P_{i}}} \odot e_{id_x}$ to be implementable, where $P_i$ is the probability mass of cluster $i$. Thus, we need Assumption 5 to make sure this happens. We apologize for a typo in the previous proof, where Assumption 5 is typed as Assumption 1. We’ve fixed it in our new revision.
> > >
> > > Alternatively, we can combine Assumption 4 & 5 and directly assume $\frac{1}{\sqrt{P_{i}}} \odot e_{id_x}$ implementable. We separate them into two assumptions to make the statements more intuitive.
> > >
> > > > Mainly because when it has limited capacity, a higher dimensional feature may contain a lot of “wrong features” while omitting the “right features" Could you explain this? This seems to suggest that the result breaks down if . There should be some flexibility here.
> > >
> > > In our theoretical framework, the optimal features are the top eigenfunctions of the graph Laplacian. Let’s say $\mu_1, \mu_2, \cdots$, are the eigenfunctions corresponding to the smallest eigenvalues of the Laplacian. To solve the downstream problem, the learned feature needs to recover $\mu_1, \cdots, \mu_m$.
> > >
> > > When the function has an infinite capacity (which is not the focus of this paper), the learned k-dimensional feature map will exactly contain $\mu_1, \cdots, \mu_k$, hence it recovers all the top eigenfunctions we need with large enough $k$ ($k>=m$).
> > >
> > > With limited capacity, when $k=m$, our assumption guarantees that some function in the function class is essentially $\mu_1, \cdots, \mu_m$. Thus, contrastive learning will learn this function regardless of what the other functions are.
> > >
> > > However, when $k>m$ and the function class has limited capacity, the learned representation may omit some top eigenfunctions. For example, let $k = m+1$, and consider a function family containing only two ($m+1$)-dimensional feature maps $f_1$ and $f_2$. The first feature map $f_1$ is composed of $\mu_1, \mu_2, \cdots, \mu_m$, and $\mu_t$ for some very large t, that is, it contains all the perfect eigenfunctions but one terrible eigenfunction ($\mu_t$). Also, suppose the second feature map $f_2$ is composed of $\mu_1 \cdots, \mu_{m-1}$ and $\mu_{m+1}, \mu_{m+2}$, i.e., it contains all the necessary eigenfunctions for solving downstream except for $\mu_m$, but also two eigenfunctions $\mu_{m+1}, \mu_{m+2}$ that are relatively good in terms of the eigenvalues or in terms of their contribution to the contrastive loss.  $f_1$ can solve the downstream problem perfectly but $\mu_t$ contributes a larger contrastive loss. On the other hand, $f_2$ achieves a smaller contrastive loss (because it doesn’t have $\mu_t$), but it omits $\mu_m$ and thus cannot achieve as good downstream performance as $f_1$. The issue is that the contrastive loss will prefer $f_2$ over $f_1$.
> > >
> > > To allow for a more flexible choice of k, we believe that additional assumptions on the structure of the family of the models need to be made to avoid the above situation. For example, if the family also contains $f_3$ that is composed of $\mu_1, \dots, \mu_{m+1}$, then $f_3$ will be chosen instead of either $f_1$ and $f_2$, and the issue is solved. So ideally if the family of models has some property that ensures that if both $f_1$ and $f_2$ are in the family and then $f_3$ is in the family, then we can allow a flexible choice of k. This is left for an interesting open question.
> > >
> > > > Do the authors have any ideas on how to get a more rigorous experimental evaluation? If not, the authors should tone down that section and its claims.
> > >
> > > Thanks for the suggestion! We’ve moved the experiments section into the appendix. We would like to emphasize that our core contribution is the theory, and experiments are only complementary.

---

> > > > ### Comment · Reviewer_1bvx · 2022-11-27
> > > > **Response to "Further Response"**
> > > >
> > > > Thanks for answering my questions. I will increase my score to 6, and lean towards acceptance. However, the brittleness of the proof with respect to the implementability assumption (Assumption 4 and 5), and the exact value of $k = m$ make me less excited about the paper. I encourage the authors to have an honest discussion of the limitations in the paper, and add the explanations they gave me to the paper, specifically on the additional assumptions needed to get rid of the brittleness of $k = m$.

---

> > > > > ### Author Response · Authors · 2022-12-01
> > > > > **Thanks for your response**
> > > > >
> > > > > We thank the reviewer for raising the score and for leaning towards acceptance of our work. We will add our discussions about Assumption 4 and 5 and the selection of feature dimension k=m into our next revision, and we'll also include more discussions regarding the limitations of our work.

---

### Official Review · Reviewer_tjye · 2022-11-03

**Confidence:** 3
**Correctness:** 2
**Technical Novelty And Significance:** 3
**Empirical Novelty And Significance:** 2
**Recommendation:** 6

**Clarity, Quality, Novelty And Reproducibility:**

Quality.

Strong theoretical results, but some of the mathematical details are not clear. Experiments are not very convincing.

Clarity.

The paper is mostly well-written, but as showed above, some of the details can be clarified.

Novelty.

The paper contains several novel theoretical results.

**Strength And Weaknesses:**

Strengths
1. Solid theoretical results.

2. Novel results for self-supervised learning with contrastive loss.

Weaknesses

The main weakness is that the presentation of the results and mathematical details are not very clear.

Here are the details:

1. The relevance of Theorem 1 in this paper is not clear. It is presented as the main result but is not used in the paper. Theorem 2 is used instead. It is claimed that Theorem 2 generalizes Theorem 1. However, because of the various assumptions, it is not shown if this generalization is strict, or there are cases where Theorem 1 guarantees better results. Generally, the connection between the assumptions in Section 3 and Section 4 are not clear.
2. The details of the proof of Theorem 2 are not clear:
(a) Where is Eq. (57) used in the proof? Also, $\tilde{W}$ appears in the proof (Eq. 57), but later disappears. Where does it appear in the calculations?
(b) How is Eq. (59) derived from Lemma 1 and Eq. (55)? The details are not shown.
3. In Example 3, what does “contains lots of disconnected subsets” mean formally?
4. In Theorem 1, does the result hold for k>m? In Theorem 3, can we set k > s?
5. The experiments are not very convincing. The optimization method that first optimizes the contrastive loss and then whitens the result may not minimize Eq. (18).
6. Are the results in Section 5 specific to the regularization used? For example, if we slightly change the distribution in Example 1 to a general product distribution, do the results still hold?
7. The orange points in Figure 1 are missing.


**Summary Of The Paper:**

This paper studies contrastive learning from a theoretical perspective. Previous works on this topic showed the benefits of contrastive learning, assuming the pretraining loss is minimized over all sets of functions with no specific form. In this work, it is assumed that the pretraining loss is minimized by certain classes of functions, such as linear predictors and neural networks. Under different assumptions on the data and architecture, it is shown that contrastive learning can learn useful representations for downstream tasks. The results show that this can be achieved with representations of lower dimensionality compared to previous results.

**Summary Of The Review:**

This paper shows novel theoretical results on self-supervised learning with the contrastive loss. In several parts, the presentation of the mathematical results and writing are not clear.

---

> ### Author Response · Authors · 2022-11-12
> **Response to Reviewer tjye**
>
> We thank the reviewer for the positive review, expressing that our work has “strong theoretical results” and is “well-written”.
>
> We want to emphasize that the core contribution of this work is to provide the **first theoretical analyses** of how inductive biases originating from the model class lead to better contrastive learned representations, shedding light on a recently observed empirical mystery.
>
> We address the comments of the reviewers below.
>
> >The relationship between Theorem 1 and Theorem 2.
>
> Both Theorem 1 and Theorem 2 should be viewed as the main results of this paper. In particular, Theorem 1 assumes that the graph is composed of several minimal implementable clusters, and shows that when the feature dimension equals the number of minimal implementable clusters, the learned feature is guaranteed to be good for downstream tasks. Theorem 2 considers a more general situation where the minimal implementable clusters may not be well-defined, yet still we can show good results when the dimensionality is equal to the number of achievable eigenfunctions. Theorem 2 generalizes Theorem 1 because, in the setting of Theorem 1, the identity function of each minimal implementable cluster would be an achievable eigenfunction. We use Theorem 2 for the examples because it’s more general and easier to be used, whereas we present Theorem 1 because it’s more intuitive to understand. For instance, in Example 1, Theorem 1 only applies when s=1, whereas Theorem 2 applies for arbitrary s.
>
> >Details about Theorem 2
>
> Thanks for pointing out the confusion in our proof — we’ve updated the proof in our new version to make it more clear.
>
> >The meaning of “contains lots of disconnected subsets” in example 3
>
> It simply means that each of the clusters **may** contain multiple disconnected subsets. Sorry for the confusion — we’ve fixed the language in our updated version.
>
> > In Theorem 1, does the result hold for k>m? In Theorem 3, can we set k > s?
>
> We agree this is a limitation of our current theorem — we need to set k equal to m (in Thm1) or s (in Thm2) for the proof to go through. This is because when we set a larger k, the minimizer of the contrastive loss may learn two larger eigenfunctions but ignore some smaller eigenfunctions. We believe this is an issue with the contrastive loss function that we study, and addressing this question may require theoretically studying other contrastive losses which is still a challenging open question.
>
> >The optimization method that first optimizes the contrastive loss and then whitens the result may not minimize Eq. (18)
>
> We agree that it’s challenging to minimize eq.(18) empirically, and our experiment is simply our best effort to achieve that. We’d like to emphasize that our main contribution is the theory, whereas the experiments are just supplementary.
>
> > If we slightly change the distribution in Example 1 to a general product distribution, do the results still hold?
>
> Yes, it still holds — our Example 1 is not dependent on the coordinate system.
>
> >The orange points in Figure 1 are missing.
>
> Sorry for the confusion, we’ve updated the figure in our new version.

---

### Author Response · Authors · 2022-11-12
**General Response**

We thank the reviewers for their thoughtful and constructive reviews of our paper. We were encouraged to hear that reviewers find that our work contains strong and novel theoretical results (tjye, 1bvx), is well-written (tjye, XWTN), has a clear motivation (XWTN, HMnY) and good structure (HMnY), hence “would be interesting to the ICLR audience” (1bvx). Reviewer 1bvx found that our result “improves the current theoretical understanding of self-supervised learning”, that we “do a good job of including several examples to instantiate the theorem”, and that “the eigenfunction perspective is a nice generalization”. Reviewer HMnY found that our work is “based on reasonable assumptions”, “introduces novel tools”, and “gets a non-trivial conclusion”.

We acknowledge that reviewers have concerns about the approximations made in the numerical experiments, especially the interpretation of the quantity b_r that we defined and empirically tested. We want to emphasize that our main contribution is the theory, and experiments are only complementary. The assumptions in our theory are challenging to verify empirically, thus we propose this experiment as our best effort to verify them. We also acknowledge that some reviewers have concerns that our writing of theoretical assumptions and exposition of the relationship between our two main theorems are not entirely clear. We’ve addressed these issues according to our individual responses, and have updated the manuscript to make the writing more clear.

We would like to reiterate the main contribution and novelty of our work:

(i) Main contribution: The core contribution of this work is to provide **theoretical analyses** of how inductive biases originating from the model class lead to better contrastive learned representations. As far as we know, we provide the **first** theoretical analysis of self-supervised learning that incorporates the effect of inductive biases, shedding light on a recently observed empirical mystery [Saunshi et al. 2022].

(ii) Novelty: We develop new proof techniques in our analysis, advancing the current status of contrastive learning theory. Our analysis relies on the novel concept of minimal implementable clusters, which clearly abstractifies the role of inductive biases in the setting of contrastive learning. We also introduce eigenfunctions as an analysis tool.

We have incorporated reviewers’ feedback, and we believe this makes our writing much more clear and our paper much stronger. In particular, we have made the following changes (and have marked the change with orange color):
1. Fixed Figure 1 to include the orange points
2. Added more explanation of the positive-pair graph and a demonstrative figure in sec 3.
3. Added one paragraph in sec 3 discussing the main contribution of this work compared with the prior work, i.e., the reduction of feature dimensionality.
4. Improved the structure of sec 3 to make the definition of partitions/downstream closer to each other, and move assumptions about the function classes later.
5. Added more discussion regarding the “closure under scaling” assumption.
6. Added a paragraph discussing the relationship between our result and [HaoChen et al 2021].
7. Added a paragraph discussing the relationship between Theorem 2 and Theorem 1.

References:

[Saunshi et al. 2022] Understanding contrastive learning requires incorporating inductive biases. ICML 2022.

[HaoChen et al. 2021] Provable Guarantees for Self-Supervised Deep Learning with Spectral Contrastive Loss. NeurIPS 2021.

---

### Decision · Program_Chairs · 2023-01-20

**Decision:**

Accept: poster

**Justification For Why Not Higher Score:**

As mentioned above, the paper does make some strong assumptions, and the empirical evaluation is almost non-existent, so I think a poster level presentation is adequate.

**Justification For Why Not Lower Score:**

The paper does make progress on an important and challenging problem.

**Metareview: Summary, Strengths And Weaknesses:**

(a) The paper presents a theoretical analysis of contrastive learning. Previous work had shown that without any assumption on architecture, such learning can fail. Thus, this paper aims to prove success of contrastive learning for a particular architecture that is in a sense aligned
with the true structure of the data.
(b) Understanding such learning settings is challenging and important, and the paper does make progress on this question, demonstrating the type of structure that contrastive learning may be good in finding.
(c) There are some strong assumptions made (e.g., that dimensionality of model fits underlying dimensionality in data) that would be good to relax. The empirical evaluation is also very minimal, which is disappointing.

**Note From Pc:**

if the above contains the word "oral" or "spotlight" please see: "oral" presentation means -> notable-top-5% and "spotlight" means -> notable-top-25%. As stated in our emails, we are disassociating presentation type from AC recommendations

**Summary Of Ac-Reviewer Meeting:**

In discussion with reviewers some questions were raised about the proof, and the authors have clarified it. As a result, all reviewers are now supportive.